# Proximal Learning With Opponent-Learning Awareness

**Stephen Zhao**
University of Toronto and Vector Institute
`stephen.zhao@mail.utoronto.ca`

**Chris Lu**
FLAIR, University of Oxford
`christopher.lu@exeter.ox.ac.uk`

**Roger Grosse**
University of Toronto and Vector Institute
`rgrosse@cs.toronto.edu`

**Jakob Foerster**
FLAIR, University of Oxford
`jakob.foerster@eng.ox.ac.uk`

## Abstract

Learning With Opponent-Learning Awareness (LOLA) (Foerster et al. [2018a]) is a multi-agent reinforcement learning algorithm that typically learns reciprocity-based cooperation in partially competitive environments. However, LOLA often fails to learn such behaviour on more complex policy spaces parameterized by neural networks, partly because the update rule is sensitive to the policy parameterization. This problem is especially pronounced in the opponent modeling setting, where the opponent's policy is unknown and must be inferred from observations; in such settings, LOLA is ill-specified because behaviourally equivalent opponent policies can result in non-equivalent updates. To address this shortcoming, we reinterpret LOLA as approximating a proximal operator, and then derive a new algorithm, proximal LOLA (POLA), which uses the proximal formulation directly. Unlike LOLA, the POLA updates are *parameterization invariant*, in the sense that when the proximal objective has a unique optimum, behaviourally equivalent policies result in behaviourally equivalent updates. We then present practical approximations to the ideal POLA update, which we evaluate in several partially competitive environments with function approximation and opponent modeling. This empirically demonstrates that POLA achieves reciprocity-based cooperation more reliably than LOLA.

## 1 Introduction

As autonomous learning agents become more integrated into society, it is increasingly important to ensure these agents' interactions produce socially beneficial outcomes, i.e. those with high total reward. One step in this direction is ensuring agents are able to navigate social dilemmas [Dawes, 1980]. Foerster et al. [2018a] showed that simple applications of independent reinforcement learning (RL) to social dilemmas usually result in suboptimal outcomes from a social welfare perspective. To address this, they introduce *Learning With Opponent-Learning Awareness* (LOLA), which actively shapes the learning step of other agents. LOLA with tabular policies learns *tit-for-tat* (TFT), i.e. reciprocity-based cooperation, in the iterated prisoner's dilemma (IPD), a 2-agent social dilemma.

However, for the same IPD setting, we show that LOLA with policies parameterized by neural networks often converges to the Pareto suboptimal equilibrium of unconditional defection. This shows that the learning outcome for LOLA is highly dependent on *policy parameterization*, one factor that makes LOLA difficult to scale to higher dimensional settings. This problem is especially pronounced in the *opponent modeling* setting, where the opponent's policy is *unknown* and must be

36th Conference on Neural Information Processing Systems (NeurIPS 2022).

inferred from observations; in such settings, LOLA is ill-specified because behaviourally equivalent opponent policies can result in non-equivalent updates and hence learning outcomes.

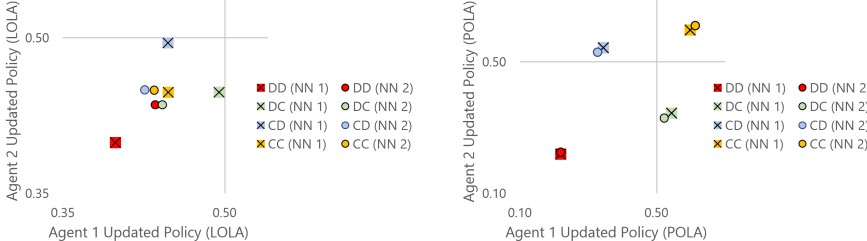

Figure 1: In the IPD (Section 4.1), we initialize two neural networks using the same architecture with different weights that produce the same policy, and plot the probability of cooperation in 4 states (DD, DC, CD, CC) after a single update of LOLA and *outer POLA* (Section 3.3). Left: using LOLA; comparing circles to crosses, the different policy parameterizations result in very different updated policies despite the same starting policies and hyperparameters. Right: using *outer POLA*; the two updated policies are much more similar.

To address these issues, we build on ideas from *proximal algorithms* [Parikh and Boyd, 2014], reinterpreting LOLA as a 1-step approximation of such a proximal operator, and then devise a new algorithm, *proximal LOLA* (POLA), which uses this proximal formulation directly. POLA replaces the gradient updates in LOLA with proximal operators based on a *proximity penalty* in policy space, which makes POLA invariant to policy parameterization. When the proximal objective has a unique optimum, behaviourally equivalent policies result in behaviourally equivalent updates.

Solving for the exact POLA update is usually intractable, so we also provide practical algorithms for settings with and without environment rollouts that rely on approximate versions of the POLA update. We show empirically that our algorithms learn reciprocity-based cooperation with significantly higher probability than LOLA in several settings with *function approximation*, even when combined with opponent modeling.

Figure 1 illustrates our core idea: LOLA is sensitive to policy parameterization, whereas POLA is not. We compare the same policy parameterized by two different sets of weights in the same neural network architecture in the IPD, showing that our approximation to the POLA update is largely parameterization independent, while the LOLA update varies depending on parameterization.

For reproducibility, our code is available at: `https://github.com/Silent-Zebra/POLA`.

## 1.1 Summary of Motivation, Contributions and Outline of Paper Structure

**Motivation:** Reciprocity-based cooperation is a desired learning outcome in partially competitive environments (Section 2.2). LOLA typically learns such behaviour in simple policy spaces, but not on more complex policy spaces parameterized by neural networks. Our goal is to more reliably learn reciprocity-based cooperation in such policy spaces.

**Contributions:**

- We identify and demonstrate a previously unknown problem with LOLA that helps explain the aforementioned issue: LOLA is sensitive to policy parameterization (Section 3.1).

- To address this sensitivity, we conceptually reinterpret LOLA as approximating a proximal operator, and derive a new algorithm, *ideal POLA*, which uses the proximal formulation directly (Section 3.2).

- *Ideal POLA* updates are invariant to policy parameterization, but usually intractable in practice. So, we develop approximations to the *ideal POLA* update for use in practice (Sections 3.3, 3.4).

- We demonstrate that our approximations more reliably learn reciprocity-based cooperation than LOLA in several partially competitive environments (Section 4.1), including with function approximation and opponent modeling (Sections 4.2, 4.3).

## 2 Background

### 2.1 Reinforcement Learning

We consider the standard fully-observable reinforcement learning and Markov Game formulation:

An $N$-agent fully observable Markov Game $\mathcal{M}$ is defined as the tuple $\mathcal{M} = \langle \mathcal{S}, \mathcal{A}, \mathcal{T}, \mathcal{R}, \gamma \rangle$, where $\mathcal{S}$ is the state space, $\mathcal{A} = \{A_1, ..., A_N\}$ is a set of action spaces, where $\mathcal{A}_i$ for $i \in \{1, ..., N\}$ denotes the action space for agent $i$, $\mathcal{T} : \mathcal{S} \times \mathcal{A}_1 \times ... \times \mathcal{A}_N \to \mathcal{P}(\mathcal{S})$ is a transition function mapping from states and actions to a probability distribution over next states, $\mathcal{R} = \{\mathcal{R}_1, ..., \mathcal{R}_N\}$ is a set of reward functions, where $\mathcal{R}_i : \mathcal{S} \times \mathcal{A}_1 \times ... \times \mathcal{A}_N \to \mathbb{R}$ denotes the reward function for agent $i$, and $\gamma \in \mathbb{R}$ is a discount factor. Each agent acts according to a policy $\pi_{\theta^i} : \mathcal{S} \to \mathcal{P}(\mathcal{A}_i)$, with parameters $\theta^i$.

Let $\boldsymbol{\theta}$ be the concatenation of each of the individual $\theta^i$ (so for $N = 2, \boldsymbol{\theta} = \{\theta^1, \theta^2\}$) and $\pi_{\boldsymbol{\theta}}$ be the concatenation of each of the individual $\pi_{\theta^i}$. Each agent's objective is to maximize its discounted expected return: $J^i(\pi_{\boldsymbol{\theta}}) = \mathbb{E}_{s \sim P(\mathcal{S}), \mathbf{a} \sim \pi_{\boldsymbol{\theta}}} \left[ \sum_{t=0}^{T} \gamma^t r_t^i \right]$ where $\mathbf{a} \sim \pi_{\boldsymbol{\theta}}$ is a joint action $\mathbf{a} = \{a_1, ..., a_N\} \in \mathcal{A}_1 \times ... \times \mathcal{A}_n$ drawn from the policies $\pi_{\boldsymbol{\theta}}$ given the current state $s \in \mathcal{S}$, $P(\mathcal{S})$ is the probability distribution over states given the current policies and transition function, $r_t^i$ denotes the reward achieved by agent $i$ at time step $t$ as defined by the reward function $\mathcal{R}_i$ and $T \in \mathbb{N}$ is the total time horizon or episode length. Throughout this paper, we assume all states are visited with non-zero probability. This is true in practice with stochastic policies, which we use throughout our experiments.

Let $L^i(\pi_{\boldsymbol{\theta}}) = -J^i(\pi_{\boldsymbol{\theta}})$. Each agent's objective is equivalently formulated as minimizing $L^i(\pi_{\boldsymbol{\theta}})$. We use this as it better aligns with standard optimization frameworks. For ease of exposition throughout this paper we consider the $N = 2$ case and consider updates from the perspective of agent 1. Updates for agent 2 follow the same structure, but with agents 1 and 2 swapped.

### 2.2 Reciprocity-Based Cooperation and the IPD

Reciprocity-based cooperation refers to cooperation *iff* others cooperate. Unlike *unconditional cooperation*, this encourages other self-interested learning agents to cooperate. One well-known example is the *tit-for-tat* (TFT) strategy in the IPD [Axelrod and Hamilton, 1981]. At each time step in the IPD, agents cooperate (C) or defect (D). Defecting always provides a higher individual reward compared to cooperating for the given time step, but both agents are better off when both cooperate than when both defect. The game is played repeatedly with a low probability of termination at each time step, which is modeled by the discount factor $\gamma$ in RL. TFT agents begin by cooperating, then cooperate *iff* the opponent cooperated at the previous time step. Against a TFT agent, the best strategy for a self-interested agent is to cooperate at every time step. Other examples of reciprocity-based cooperation include contributing to a common good when others contribute (to avoid punishment), or reciprocating by collecting only resources that do not harm others (such as in Section 4.3). Crucially, *unconditional cooperation*, for example in the single step prisoner's dilemma or against defecting opponents in the IPD, is usually a *dominated strategy* and thus not a desired learning outcome. Thus, we do not compare against works that modify the learning objective and can in principle learn *unconditional cooperation* such as Hughes et al. [2018], Wang et al. [2018], Jaques et al. [2019], McKee et al. [2020].

### 2.3 Learning with Opponent-Learning Awareness (LOLA)

LOLA [Foerster et al., 2018a] introduces *opponent shaping* via a gradient based approach. Instead of optimizing for $L^1(\pi_{\theta^1}, \pi_{\theta^2})$, i.e. the loss for agent 1 given the policy parameters $(\theta^1, \theta^2)$, agent 1 optimizes for $L^1(\pi_{\theta^1}, \pi_{\theta^2 - \Delta\theta^2(\theta^1)})$, the loss for agent 1 after agent 2 updates its policy with one naive learning (NL) gradient step $\Delta\theta^2(\theta^1) = \eta\nabla_{\theta^2}L^2(\pi_{\theta^1}, \pi_{\theta^2})$, where $\eta$ is the opponent's learning rate. Importantly, $\Delta\theta^2$ is treated as a function of $\theta^1$, and LOLA differentiates through the update $\Delta\theta^2(\theta^1)$ when agent 1 optimizes for $L^1(\pi_{\theta^1}, \pi_{\theta^2})$. Appendix A.1 provides more details, including pseudo-code for LOLA. Foerster et al. [2018a] also provide a policy gradient based formulation for use with environment rollouts when the loss cannot be calculated analytically, and use opponent modeling to avoid needing access to the opponent's policy parameters.

In the IPD with tabular policies, LOLA agents learn a TFT strategy, with high probability cooperating when the other agent cooperates and defecting when the other defects [Foerster et al., 2018a]. Thus, LOLA showed that *with the appropriate learning rule* self-interested agents can learn policies that lead to socially optimal outcomes in the IPD.

## 2.4 Proximal Point Method

Following Parikh and Boyd [2014], define the proximal operator $\text{prox}_{\lambda f} : \mathbb{R}^n \to \mathbb{R}^n$ of a closed proper convex function $f$ as:

$$\text{prox}_{\lambda f}(v) = \arg\min_x \left( f(x) + \frac{1}{2\lambda} ||x - v||_2^2 \right)$$

where $|| \cdot ||_2$ is the Euclidean (L2) norm and $\lambda$ is a hyperparameter. If $f$ is differentiable, its first-order approximation near $v$ is:

$$\hat{f}_v^{(1)}(x) = f(v) + \nabla f(v)^T (x - v)$$

The proximal operator of the first-order approximation is:

$$\text{prox}_{\hat{f}_v^{(1)}}(v) = \arg\min_x \left( f(v) + \nabla f(v)^T (x - v) + \frac{1}{2\lambda} ||x - v||_2^2 \right) = v - \lambda \nabla f(v)$$

which is a standard gradient step on the original function $f(v)$ with step size $\lambda$.

The proximal point method starts with an iterate $x_0$, then for each time step $t \in \{1, 2, ...\}$, calculates a new iterate $x_t = \text{prox}_{\lambda f}(x_{t-1})$. Gradient descent is equivalent to using the proximal point method with a first-order approximation of $f$.

# 3 Proximal LOLA (POLA)

In this section, we first explore how LOLA is sensitive to different types of changes in policy parameterization (Section 3.1). Next, we introduce *ideal POLA* (Section 3.2), a method invariant to all such changes. Lastly, we present approximations to POLA and resulting practical algorithms (Sections 3.3, 3.4).

## 3.1 Sensitivity to Policy Parameterization

LOLA is sensitive to policy parameterization, as it is defined in terms of (Euclidean) gradients, which are not invariant to smooth transformations of the parameters. Specifically, parameterization affects not only the convergence rate, but also which equilibrium is reached, as we illustrate with a simple toy example. Consider the case of tabular policies with one time step of memory for the IPD. There are five possible states: DD (both agents last defected), DC (agent 1 defected while agent 2 cooperated), CD (agent 1 cooperated, agent 2 defected), CC (both agents last cooperated), and the starting state. For each agent $i \in \{1, 2\}$, $\theta^i \in \mathbb{R}^5$ parameterizes a tabular policy over these 5 states, so that $\pi_{\theta^i}(s) = \text{sigmoid}(\theta^i)[s]$, where $v[s]$ denotes the element of vector $v$ corresponding to state $s$. To illustrate the dependence on parameterization, we apply the invertible transformation $\mathbf{Q}^i \theta^i$ where:

$$\mathbf{Q}^1 = \begin{pmatrix} 1 & 0 & -2 & 0 & 0 \\ 0 & 1 & -2 & 0 & 0 \\ 0 & 0 & 1 & 0 & 0 \\ 0 & 0 & -2 & 1 & 0 \\ 0 & 0 & -2 & 0 & 1 \end{pmatrix}, \mathbf{Q}^2 = \begin{pmatrix} 1 & -2 & 0 & 0 & 0 \\ 0 & 1 & 0 & 0 & 0 \\ 0 & -2 & 1 & 0 & 0 \\ 0 & -2 & 0 & 1 & 0 \\ 0 & -2 & 0 & 0 & 1 \end{pmatrix}$$

Thus, agent $i$'s policy is now $\pi_{\mathbf{Q}^i \theta^i}(s) = \text{sigmoid}(\mathbf{Q}^i \theta^i)[s]$.

**Definition 3.1.** For policies $\pi_{\theta^{ia}}, \pi_{\theta^{ib}}$, we say $\pi_{\theta^{ia}} = \pi_{\theta^{ib}}$ when for all $s \in \mathcal{S}$, $\pi_{\theta^{ia}}(s) = \pi_{\theta^{ib}}(s)$.

The transformation matrices $\mathbf{Q}^1, \mathbf{Q}^2$ are non-singular and represent changes of basis. Thus, for any policy $\pi_{\theta^i}$, there exists some $\theta^{i\prime}$ such that $\pi_{\mathbf{Q}^i \theta^{i\prime}} = \pi_{\theta^i}$. Despite this, LOLA fails to learn TFT in this transformed policy space (Figure 3). We also observe this issue when comparing tabular LOLA to LOLA with policies parameterized by neural networks in Figure 3; changes in policy representation materially affect LOLA. Relatedly, LOLA is sensitive to different policy parameterizations with the same policy representation, as Figure 1 (left) shows. In the next subsection, we propose an algorithm that addresses these issues.

## 3.2 Ideal POLA

In this section, we first formalize the desired invariance property. Next, we introduce an idealized (but impractical) version of POLA, and show that it achieves the desired invariance property.

**Definition 3.2.** An update rule $u : \mathbb{R}^n \times \mathbb{R}^m \to \mathbb{R}^n$ that updates policy parameters $\theta^1 \in \mathbb{R}^n$ using auxiliary information $\theta^2 \in \mathbb{R}^m$ is invariant to policy parameterization when for any $\theta^{1a} \in \mathbb{R}^n, \theta^{1b} \in \mathbb{R}^n, \theta^{2a} \in \mathbb{R}^m, \theta^{2b} \in \mathbb{R}^m$ such that $\pi_{\theta^{1a}} = \pi_{\theta^{1b}}$ and $\pi_{\theta^{2a}} = \pi_{\theta^{2b}}$, $\pi_{u(\theta^{1a}, \theta^{2a})} = \pi_{u(\theta^{1b}, \theta^{2b})}$. In short, if the original policies were the same, so are the new policies (but not necessarily the new policy parameters).

To achieve invariance to policy parameterization, we introduce our idealized version of proximal LOLA (*ideal POLA*). Similarly to PPO [Schulman et al., 2017], each player adjusts their policy to increase the probability of highly rewarded states while penalizing the distance moved in policy space. Crucially, each player also assumes the other player updates their parameters through such a proximal update. More formally, at each policy update, agent 1 solves for the following $\theta^{1\prime}$:

$$\theta^{1\prime}(\theta^1, \theta^2) = \arg\min_{\theta^{1\prime\prime}} \left( L^1(\pi_{\theta^{1\prime\prime}}, \pi_{\theta^{2\prime}(\theta^{1\prime\prime}, \theta^2)}) + \beta_{\text{out}} D(\pi_{\theta^1} || \pi_{\theta^{1\prime\prime}}) \right) \tag{1}$$

where $D(\pi_{\theta^i} || \pi_{\theta^{i\prime\prime}})$ is shorthand for a general divergence defined on policies; specific choices are given in subsequent sections. For $\theta^{2\prime}$ in Equation 1, we choose the following proximal update:

$$\theta^{2\prime}(\theta^{1\prime\prime}, \theta^2) = \arg\min_{\theta^{2\prime\prime}} \left( L^2(\pi_{\theta^{1\prime\prime}}, \pi_{\theta^{2\prime\prime}}) + \beta_{\text{in}} D(\pi_{\theta^2} || \pi_{\theta^{2\prime\prime}}) \right) \tag{2}$$

For the above equations to be well defined, the $\arg\min$ must be unique; we assume this in all our theoretical analysis. If different parameters can produce the same policy, or multiple different policies have the same total divergence and loss, then the $\arg\min$ is non-unique. However, non-unique solutions is not an issue in practice, as our algorithms approximate $\arg\min$ operations with multiple gradient updates.

**Theorem 3.3.** *The POLA update rule $u(\theta^1, \theta^2) = \theta^{1\prime}(\theta^1, \theta^2)$, where $\theta^{1\prime}$ is defined based on Equations 1 and 2, is invariant to policy parameterization.*

The proof is in Appendix A.2. In short, this follows from the fact that all terms in the $\arg\min$ in Equations 1 and 2 are functions of policies, and not directly dependent on policy parameters.

This invariance also is a major advantage in opponent modeling settings, where agents cannot directly access the policy parameters of other agents, and must infer policies from observations. LOLA is ill-specified in such settings; the LOLA update varies depending on the assumed parameterization of opponents' policies. Conversely, POLA updates are invariant to policy parameterization, so any parameterization can be chosen for the opponent model.

To clarify the relation between LOLA and POLA, LOLA is a version of POLA that uses first-order approximations to all objectives and an L2 penalty on policy parameters rather than a divergence over policies. This follows from gradient descent being equivalent to the proximal point method with first order approximations (Section 2.4); we provide a full proof in Appendix A.3.

Exactly solving Equations 1 and 2 is usually intractable, so in the following Sections 3.3 and 3.4, we formulate practical algorithms that approximate the POLA update.

## 3.3 Outer POLA

To approximate *ideal POLA*, we use a first order approximation to agent 2's objective, which is equivalent to agent 2 taking a gradient step (see Theorem A.3 for details). That is, instead of finding $\theta^{2\prime}$ via Equation 2, we use $\theta^{2\prime} = \theta^2 - \Delta\theta^2$. Agent 1 thus solves for:

$$\theta^{1\prime}(\theta^1, \theta^2) = \arg\min_{\theta^{1\prime\prime}} \left( L^1(\pi_{\theta^{1\prime\prime}}, \pi_{\theta^2 - \Delta\theta^2(\theta^{1\prime\prime})}) + \beta_{\text{out}} D(\pi_{\theta^1} || \pi_{\theta^{1\prime\prime}}) \right) \tag{3}$$

Solving the $\arg\min$ above exactly is usually intractable. For a practical algorithm, we differentiate through agent 2's gradient steps with unrolling, like in LOLA, and repeatedly iterate with gradient steps on agent 1's proximal objective until a fixed point is found; Algorithm 1 shows pseudo-code. We choose $D(\pi_{\theta^1} || \pi_{\theta^{1\prime\prime}}) = \mathbb{E}_{s \sim U(\mathcal{S})}[D_{KL}(\pi_{\theta^1}(s) || \pi_{\theta^{1\prime\prime}}(s))]$, where $U(\mathcal{S})$ denotes a uniform distribution over states, as this is most analogous to an L2 distance on tabular policies, and we only test *outer POLA* on settings where tabular policies can be used.

---
**Algorithm 1** Outer POLA 2-agent formulation: update for agent 1
---

**Input:** Policy parameters $\theta^1, \theta^2$, proximal step size $\alpha_1$, learning rate $\eta$, penalty strength $\beta_{\text{out}}$
Make copy: $\theta^{1''} \leftarrow \theta^1$
**repeat**
    $\theta^{2''} \leftarrow \theta^2 - \eta \nabla_{\theta^2} L^2(\pi_{\theta^{1''}}, \pi_{\theta^2})$
    $\theta^{1''} \leftarrow \theta^{1''} - \alpha_1 \nabla_{\theta^{1''}}(L^1(\pi_{\theta^{1''}}, \pi_{\theta^{2''}}) + \beta_{\text{out}}(\mathbb{E}_{s \sim U(\mathcal{S})}[D_{KL}(\pi_{\theta^1}(s)||\pi_{\theta^{1''}}(s))]))$
**until** $\theta^{1''}$ has converged to a fixed point
**Output:** $\theta^{1''}$

---

## 3.4 POLA-DiCE

*Outer POLA* assumes we can calculate the exact loss, but often we need to estimate the loss with samples from environment rollouts. For these cases, we introduce a policy gradient version of POLA adapted to work with DiCE [Foerster et al., 2018b]. DiCE is a sample-based estimator that makes it easy to estimate higher order derivatives using backprop. More details about DiCE and its combination with LOLA are in the Appendix (A.4, A.5) and in Foerster et al. [2018b].

POLA-DiCE approximates the $\arg\min$ in *ideal POLA* (Equations 1 and 2) with a fixed number of gradient steps on the proximal objectives, where steps on the outer objective (1) differentiate through the unrolled steps on the inner objective (2). We choose $D(\pi_{\theta^i}||\pi_{\theta^{i''}}) = \mathbb{E}_{s \sim \mathcal{S}}[D_{KL}(\pi_{\theta^i}(s)||\pi_{\theta^{i''}}(s))]$, and approximate the expectation under the true state visitation with an average over states visited during rollouts. $s_{\leq T}$ denotes the states from a set of rollouts with $T$ time steps: $s_{\leq T} \triangleq \{s_t : t \in \{1, ..., T\}\}$, where $s_t$ is the state at time step $t$. $D(\pi_{\theta^i}, \pi_{\theta^{i''}}|s_{\leq T}) \triangleq \frac{1}{|s_{\leq T}|} \sum_{s \in s_{\leq T}} [D_{KL}(\pi_{\theta^i}(s)||\pi_{\theta^{i''}}(s))]$ denotes our sample based approximation to $\mathbb{E}_{s \sim \mathcal{S}}[D_{KL}(\pi_{\theta^i}(s)||\pi_{\theta^{i''}}(s))]$. As in LOLA-DiCE, rollouts are done in a simulator, and used in the DiCE loss $\mathcal{L}^i_{\odot(\pi_{\theta^1}, \pi_{\theta^2})}$. For our experiments, we assume full knowledge of transition dynamics, but in principle an environment model could be used instead. Algorithm 2 provides pseudo-code for POLA-DiCE.

---
**Algorithm 2** POLA-DiCE 2-agent formulation: update for agent 1
---

**Input:** Policy parameters $\theta^1, \theta^2$, learning rates $\alpha_1, \alpha_2$, penalty hyperparameters $\beta_{\text{in}}, \beta_{\text{out}}$, number of outer steps $M$ and inner steps $K$
Initialize: $\theta^{1''} \leftarrow \theta^1$
**for** $m$ in $1...M$ **do**
    Initialize: $\theta^{2''} \leftarrow \theta^2$
    **for** $k$ in $1...K$ **do**
        Rollout trajectories with states $s^{\text{in}}_{\leq T}$ using policies $(\pi_{\theta^{1''}}, \pi_{\theta^{2''}})$
        $\theta^{2''} \leftarrow \theta^{2''} - \alpha_2 \nabla_{\theta^{2''}}(\mathcal{L}^2_{\odot(\pi_{\theta^{1''}}, \pi_{\theta^{2''}})} + \beta_{\text{in}} D(\pi_{\theta^2}, \pi_{\theta^{2''}}|s^{\text{in}}_{\leq T}))$
    **end for**
    Rollout trajectories with states $s^{\text{out}}_{\leq T}$ using policies $(\pi_{\theta^{1''}}, \pi_{\theta^{2''}})$
    $\theta^{1''} \leftarrow \theta^{1''} - \alpha_1 \nabla_{\theta^{1''}}(\mathcal{L}^1_{\odot(\pi_{\theta^{1''}}, \pi_{\theta^{2''}})} + \beta_{\text{out}} D(\pi_{\theta^1}, \pi_{\theta^{1''}}|s^{\text{out}}_{\leq T}))$
**end for**
**Output:** $\theta^{1''}$

---

For sufficiently large numbers of inner steps $K$ and outer steps $M$, and sufficiently small learning rates, POLA-DiCE iterates until a fixed point is found. Unfortunately, iterating to convergence often requires a very large amount of rollouts (and memory for the inner steps). When $M = 1$ and $\beta_{\text{in}}, \beta_{\text{out}} = 0$, POLA-DiCE is equivalent to LOLA-DiCE [Foerster et al., 2018b].

Without opponent modeling, LOLA-DiCE and POLA-DiCE directly access the other agent's policy parameters, using those in simulator rollouts for policy updates. In the opponent modeling (OM) setting, agents can only access actions taken by the other agent, from real environment rollouts. For policy updates, agents must use learned policy models in simulator rollouts. We learn policy models by treating observed actions as targets in a supervised classification setting, as in behaviour cloning

[Ross et al., 2011]. Figure 2 illustrates the process at each time step; we keep policy models from previous iterations and update incrementally on newly observed actions.

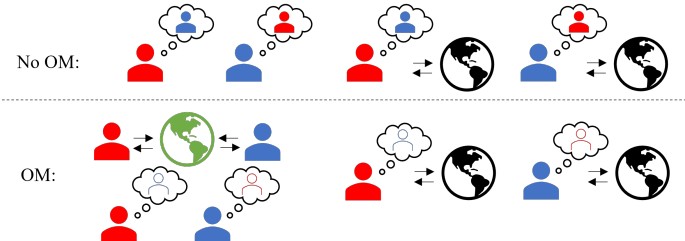

Figure 2: Illustration of the training process at each time step for LOLA-DiCE and POLA-DiCE. Without opponent modeling, each agent directly uses the other agent's policy parameters in simulator rollouts for policy updates. With opponent modeling, each agent first learns a policy model of the opponent based on observed actions from real environment rollouts. Agents then use the learned models in simulator rollouts for policy updates. We assume known dynamics, so no environment model needs to be learned for simulator rollouts.

## 4 Experiments

To investigate how useful our approximations to *ideal POLA* are, we run experiments to answer the following questions: 1) Does *outer POLA* learn reciprocity-based cooperation more consistently than LOLA, across a variety of policy parameterizations, in the IPD with one-step memory (Section 4.1)? 2) Does POLA-DiCE learn reciprocity-based cooperation more consistently than LOLA-DiCE in settings that require function approximation and rollouts (IPD with full history (Section 4.2) and coin game (Section 4.3))? If so, do these results still hold when using opponent modeling?

### 4.1 One-Step Memory IPD

With one-step memory, the observations are the actions by both agents at the previous time step. In this setting, we can use the exact loss in gradient updates (see Appendix B.1.1 for details).

To investigate behaviour across settings that vary the difficulty of learning reciprocity-based cooperation, we introduce a *cooperation factor* $f \in \mathbb{R}$, which determines the reward for cooperating relative to defecting. At each time step, let $c$ be the number of agents who cooperated. Each agent's reward is $c * f/2 - \mathbb{1}[\text{agent contributed}]$. This is a specific instance of the *contribution game* from Barbosa et al. [2020], with two agents. $1 < f < 2$ satisfies social dilemma characteristics, where defecting always provides higher individual reward, but two cooperators are both better off than two defectors. For more details on the problem setup, see Appendices B.1.2 and B.1.3.

Figure 3 compares LOLA and *outer POLA* (Section 3.3) using various $f$ and policy parameterizations. To provide a succinct graphical representation of how well agents learn reciprocity-based cooperation, we test how often agents *find TFT*. We consider agents cooperating with each other with high probability, achieving average reward $> 80\%$ of the socially optimal, but both cooperating with probability $< 0.65$ if the opponent last defected, to have *found TFT*. These thresholds are somewhat arbitrary; Appendix B.1.5 provides detailed policy probabilities that support our conclusions without such thresholds.

We reproduce Foerster et al. [2018a]'s result that naive learning converges to unconditional defection, while LOLA using tabular policies finds TFT. However, changes in policy parameterization greatly hinder LOLA's ability to find TFT. In contrast, *outer POLA* finds TFT even with function approximation, and finds TFT significantly more often than LOLA in the pre-conditioned setting described in Section 3.1. Appendix B.1.4 further discusses hyperparameter settings.

Qualitatively, Figure 1 shows that in the IPD with one-step memory, $f = 1.33$, and neural network parameterized policies, *outer POLA* closely approximates the invariance provided by *ideal POLA*.

To demonstrate that POLA allows for any choice of opponent model, Appendix B.1.6 provides further experiments in the IPD with opponent modeling, using a version of POLA similar to POLA-DiCE.

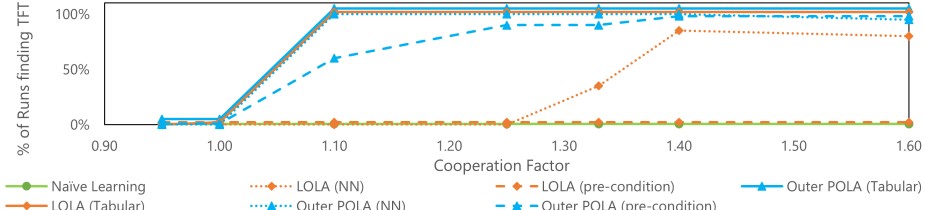

Figure 3: Comparison of LOLA and *outer POLA* in the one-step memory IPD with various $f$, plotting the percentage of 20 runs that *find TFT*. "NN" denotes policies parameterized by neural networks. "Pre-condition" denotes the setting in Section 3.1. Given hyperparameter tuning, tabular LOLA always finds TFT (for $f > 1$), whereas LOLA with function approximation fails on lower $f$. In the pre-conditioned setting, LOLA completely fails to find TFT. In contrast, *outer POLA* finds TFT regardless of whether the policy is tabular or a neural network, and retains most of its performance in the pre-conditioned setting. As sanity checks, naive learning ($\eta = 0$) always fails to find TFT, and all algorithms always defect for $f < 1$.

## 4.2 Full History IPD

Next, we relax the assumption that agents are limited to one-step memory and instead consider policies that condition on the entire history, which makes using function approximation and rollouts necessary. We parameterize policies using GRUs [Cho et al., 2014] and test whether POLA-DiCE still learns reciprocity-based cooperation within this much larger policy space. We use $f = 1.33$ for the reward structure. For more details on the problem setting, policy parameterization, and hyperparameters, see Appendix B.2.

Figure 4 shows results in this setting. LOLA agents sometimes learn reciprocity-based cooperation but often learn to always defect, achieving low scores against each other on average. POLA agents learn reciprocity-based cooperation much more consistently, almost always cooperating with each other but defecting with high probability if the opponent always defects. Furthermore, opponent modeling works well with POLA; POLA agents behave similarly using opponent modeling instead of accessing policy parameters directly.

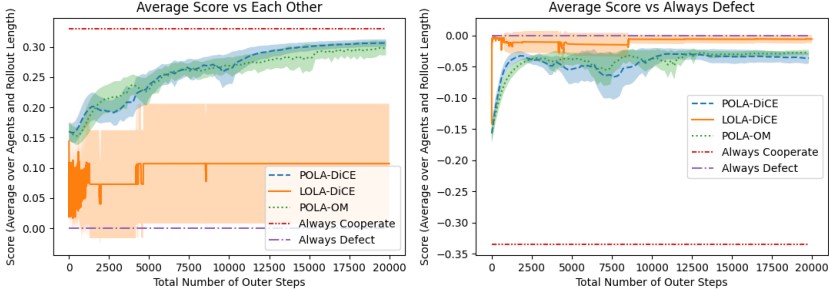

Figure 4: Comparison of LOLA-DiCE and POLA-DiCE on the IPD with GRU parameterized policies that condition on the full action history. Left: POLA-DiCE agents learn to cooperate with each other with high probability, achieving close to the socially optimal reward (always cooperate), whereas LOLA agents cooperate with each other much less consistently, often learning to always defect. Right: we test the learned policies against a hard-coded rule that always defects. POLA-DiCE agents defect against such an opponent with high probability, achieving close to the optimal reward of 0, showing POLA agents cooperate only when the other agent reciprocates. POLA-OM (POLA-DiCE with opponent modeling) agents show similar behaviour to POLA-DiCE agents. All results are averaged over 10 random seeds with 95% confidence intervals shown.

## 4.3 Coin Game

To investigate the scalability of POLA-DiCE under higher dimensional observations and function approximation, we test LOLA-DiCE and POLA-DiCE in the coin game setting from Lerer and Peysakhovich [2017]. The coin game consists of a 3x3 grid in which two agents, red and blue, move around and collect coins. There is always one coin, coloured either red or blue, which spawns with

the other colour after being collected. Collecting any coin grants a reward of 1; collecting a coin with colour different from the collecting agent gives the other agent -2 reward. The coin game embeds a temporally extended social dilemma; if agents defect by trying to pick up all coins, both agents get 0 total reward in expectation, whereas if agents cooperate by only picking up coins of their own colour, they achieve positive expected average reward (maximum: $\frac{1}{3}$ per time step). Appendix B.3 provides more detail. We again parameterize the agents' policies with GRUs [Cho et al., 2014].

Figure 5 shows that POLA-DiCE agents learn to cooperate with higher probability than LOLA-DiCE agents,[1] picking up a larger proportion of their own coins, and again do not naively cooperate. POLA-DiCE agents with opponent modeling also learn reciprocity-based cooperation, but slightly less so than with direct access to policy parameters, likely due to the noise from opponent modeling.

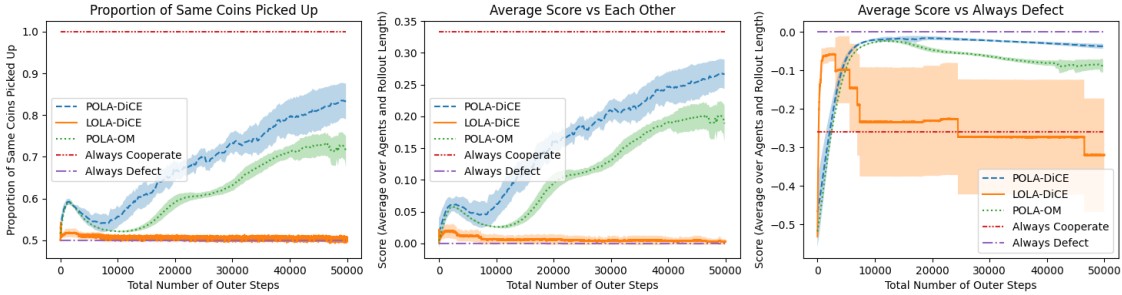

Figure 5: Comparison of LOLA-DiCE and POLA-DiCE on the coin game. Left: number of coins collected where the coin's colour matched the agent's colour, divided by the total number of coins collected. POLA agents cooperate more than LOLA agents, collecting a larger proportion of their own coloured coins. Middle: average score for the agents against each other. POLA-DiCE agents achieve close to the maximum cooperative reward. Right: we test the learned policies against a hard-coded rule that always *defects* by taking the shortest path to the coin regardless of colour. POLA agents defect, achieving scores close to the optimal of 0 against such an opponent, again showing POLA agents cooperate based on reciprocity. LOLA agents quickly learn to always defect; subsequent training is highly unstable.

## 5   Related Work

POLA-DiCE (Section 3.4) can alternatively be motivated as replacing the policy gradient update inherent in LOLA-DiCE with a proximal-style update like that of TRPO [Schulman et al., 2015a] or the adaptive KL-penalty version of PPO [Schulman et al., 2017].

There are other extensions to LOLA that address problems orthogonal to policy parameterization sensitivity. In concurrent work, Willi et al. [2022] highlight that LOLA is inconsistent in self-play; the update each player assumes the opponent makes is different from their actual update. They propose COLA, learning update rules $\Delta\theta^1$ and $\Delta\theta^2$ that are consistent with each other. However, their notion of consistency is gradient-based and thus not invariant to policy parameterization, whereas POLA is not consistent. SOS [Letcher et al., 2018], which combines the advantages of LOLA and LookAhead [Zhang and Lesser, 2010], is also not invariant to policy parameterization.

Al-Shedivat et al. [2018] formulate a meta-RL framework with a policy gradient theorem, and apply PPO as their optimization algorithm; this is one instance of combining proximal style operators with reinforcement learning style policy gradients for optimization with higher order gradients. However, they focus on adaptation rather than explicitly modeling and shaping opponent behaviour.

MATRL [Wen et al., 2021] sets up a metagame between the original policies and new policies calculated with independent TRPO updates for each agent, solving for a Nash equilibrium in the metagame at each step. MATRL finds a best response without actively shaping its opponents' learning, so it is similar to LookAhead [Zhang and Lesser, 2010] (which results in unconditional defection) rather than LOLA.

M-FOS [Lu et al., 2022] sets up a metagame where each meta-step is a full environment rollout, and the meta-reward at each step is the cumulative discounted return of the rollout. They use model-free

---

[1]See Appendix B.3 for why the LOLA results cannot be directly compared with Foerster et al. [2018a]

optimization methods to learn meta-policies that shape the opponent's behaviour over meta-steps. We do not compare against it as it is concurrent work without published code at the time of writing.

Badjatiya et al. [2021] introduce *status-quo loss*, in which agents imagine the current situation being repeated for several steps; combined with the usual RL objective, this leads to cooperation in social dilemmas. However, their method learns to defect in the one-step memory IPD in the states DC and CD, and thus does not learn reciprocity-based cooperation.

Zhao et al. [2021] provide a detailed numerical analysis of the function approximation setting for two-player zero-sum Markov Games. Fiez et al. [2021] considers a proximal method for minimax optimization. In contrast to both, we consider the general-sum setting.

## 6 Limitations

*Outer POLA* and POLA-DiCE approximate *ideal POLA*, so are not completely invariant to policy parameterization; Figures 1 and 3 show how close the approximation is in the one-step memory IPD. POLA introduces additional hyperparameters compared to LOLA (e.g. $\beta_{\text{in}}, \beta_{\text{out}}$ and number of outer steps for POLA-DiCE). POLA usually requires many rollouts for each update; future work could explore ways to mitigate this. For example, in Appendix A.8 we formulate a version of POLA-DiCE closer to PPO [Schulman et al., 2017], that repeatedly trains on samples for improved sample efficiency.

## 7 Conclusion and Future Work

Motivated by making LOLA invariant to policy parameterization, we introduced *ideal POLA*. Empirically, we demonstrated that practical approximations to *ideal POLA* learn reciprocity-based cooperation significantly more often than LOLA in several social dilemma settings.

Future work could explore and test: policy parameterization invariant versions of LOLA extensions like COLA [Willi et al., 2022] and SOS [Letcher et al., 2018]; POLA formulations with improved sample efficiency (e.g. Appendix A.8); the $N$-agent version of POLA (Appendices A.9, A.10) in $N$-agent versions of the IPD [Barbosa et al., 2020] and larger environments such as Harvest and Cleanup [Hughes et al., 2018]; POLA formulations with adaptive penalty parameters $\beta_{\text{in}}, \beta_{\text{out}}$; connections to Mirror Learning [Kuba et al., 2022]; adapting approximations to proximal point methods such as the extra-gradient method [Mokhtari et al., 2020] to work with POLA.

We are excited and hopeful that this work opens the door to scaling LOLA and related algorithms to practical settings that require function approximation and opponent modeling.

## Acknowledgements

Resources used in this research were provided, in part, by the Province of Ontario, the Government of Canada, and companies sponsoring the Vector Institute. The Foerster Lab for AI Research (FLAIR) is grateful to the Cooperative AI Foundation for their support. We thank the anonymous NeurIPS reviewers for helpful comments on earlier versions of this paper.

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
