# A   Appendix: Additional Background, Derivations, and Algorithm Details

## A.1   LOLA with Direct Update

Throughout this paper, we interpret LOLA as directly differentiating through $L(\pi_{\theta^1}, \pi_{\theta^2 - \Delta\theta^2(\theta^1)})$; we provide an algorithm box in Algorithm 3. We use this formulation of LOLA since it avoids using a first-order approximation around the objective as in Foerster et al. [2018a], and forms the conceptual basis for LOLA-DiCE [Foerster et al., 2018b]. In our experience, we find this formulation generally provides better and more consistent results, and is easier to implement.

---

**Algorithm 3** LOLA direct update 2-agent formulation: exact gradient update for agent 1

---

**Input:** Policy parameters $\boldsymbol{\theta} = \{\theta^1, \theta^2\}$, learning rates $\alpha$, $\eta$
Initialize: $\boldsymbol{\theta}' \leftarrow \boldsymbol{\theta}$
$\theta^{2\prime} \leftarrow \theta^{2\prime} - \eta\nabla_{\theta^{2\prime}}L^2(\pi_{\boldsymbol{\theta}'})$ // Preserve Computation Graph
$\theta^{1\prime} \leftarrow \theta^1 - \alpha\nabla_{\theta'_1}L^1(\pi_{\boldsymbol{\theta}'})$ // Differentiate through agent 2's update
**Output:** $\theta^{1\prime}$

---

## A.2   POLA Invariance Proof

*Proof.* Consider arbitrary $(\theta^{1a}, \theta^{1b}, \theta^{2a}, \theta^{2b})$ such that $\pi_{\theta^{1a}} = \pi_{\theta^{1b}}$ and $\pi_{\theta^{2a}} = \pi_{\theta^{2b}}$, and consider $u(\theta^{1a}, \theta^{2a})$ and $u(\theta^{1b}, \theta^{2b})$.

$$u(\theta^{1a}, \theta^{2a}) = \arg\min_{\theta^{1\prime\prime}} \left( L^1(\pi_{\theta^{1\prime\prime}}, \pi_{\theta^{2\prime}(\theta^{1\prime\prime}, \theta^{2a})}) + \beta_{\text{out}}D(\pi_{\theta^{1a}}||\pi_{\theta^{1\prime\prime}}) \right)$$

$$u(\theta^{1b}, \theta^{2b}) = \arg\min_{\theta^{1\prime\prime}} \left( L^1(\pi_{\theta^{1\prime\prime}}, \pi_{\theta^{2\prime}(\theta^{1\prime\prime}, \theta^{2b})}) + \beta_{\text{out}}D(\pi_{\theta^{1b}}||\pi_{\theta^{1\prime\prime}}) \right)$$

Recall again that we assume all $\arg\min$ to be unique. From Equation 2:

$$\theta^{2\prime}(\theta^{1\prime\prime}, \theta^{2a}) = \arg\min_{\theta^{2\prime\prime}} \left( L^2(\pi_{\theta^{1\prime\prime}}, \pi_{\theta^{2\prime\prime}}) + \beta_{\text{in}}D(\pi_{\theta^{2a}}||\pi_{\theta^{2\prime\prime}}) \right)$$

$$\theta^{2\prime}(\theta^{1\prime\prime}, \theta^{2b}) = \arg\min_{\theta^{2\prime\prime}} \left( L^2(\pi_{\theta^{1\prime\prime}}, \pi_{\theta^{2\prime\prime}}) + \beta_{\text{in}}D(\pi_{\theta^{2b}}||\pi_{\theta^{2\prime\prime}}) \right)$$

Since $\pi_{\theta^{2a}} = \pi_{\theta^{2b}}$, we get $\pi_{\theta^{2\prime}(\theta^{1\prime\prime}, \theta^{2a})} = \pi_{\theta^{2\prime}(\theta^{1\prime\prime}, \theta^{2b})}$ because:

$$\arg\min_{\theta^{2\prime\prime}} \left( L^2(\pi_{\theta^{1\prime\prime}}, \pi_{\theta^{2\prime\prime}}) + \beta_{\text{in}}D(\pi_{\theta^{2a}}||\pi_{\theta^{2\prime\prime}}) \right) = \arg\min_{\theta^{2\prime\prime}} \left( L^2(\pi_{\theta^{1\prime\prime}}, \pi_{\theta^{2\prime\prime}}) + \beta_{\text{in}}D(\pi_{\theta^{2b}}||\pi_{\theta^{2\prime\prime}}) \right)$$

Combining this with $\pi_{\theta^{1a}} = \pi_{\theta^{1b}}$, we get $u(\theta^{1a}, \theta^{2a}) = u(\theta^{1b}, \theta^{2b})$ because:

$$\arg\min_{\theta^{1\prime\prime}} \left( L^1(\pi_{\theta^{1\prime\prime}}, \pi_{\theta^{2\prime}(\theta^{1\prime\prime}, \theta^{2a})}) + \beta_{\text{out}}D(\pi_{\theta^{1a}}||\pi_{\theta^{1\prime\prime}}) \right) =$$
$$\arg\min_{\theta^{1\prime\prime}} \left( L^1(\pi_{\theta^{1\prime\prime}}, \pi_{\theta^{2\prime}(\theta^{1\prime\prime}, \theta^{2b})}) + \beta_{\text{out}}D(\pi_{\theta^{1b}}||\pi_{\theta^{1\prime\prime}}) \right)$$

Thus, $\pi_{u(\theta^{1a}, \theta^{2a})} = \pi_{u(\theta^{1b}, \theta^{2b})}$, so $u$ is invariant to policy parameterization. $\qquad\square$

## A.3   Proof of Connection Between POLA and LOLA

Here we show that LOLA is a version of POLA that uses first-order approximations to all objectives and an L2 penalty on policy parameters rather than a divergence over policies. We assume the loss is differentiable so first-order approximations are well-defined, and that all $\arg\min$ are unique.

**Lemma A.1.** *LOLA is equivalent to applying the proximal operator on a first-order approximation (around $\theta^1$) of the modified objective $L^1(\pi_{\theta^1}, \pi_{\theta^2 - \Delta\theta^2(\theta^1)})$.*

*Proof.* Let $f^1(\theta^1) = L^1(\pi_{\theta^1}, \pi_{\theta^2 - \Delta\theta^2(\theta^1)})$.

Consider the first-order Taylor approximation $\hat{f}^{1(1)}_{\theta^1}$ of the modified objective function (around $\theta^1$):[2]

$$\hat{f}^{1(1)}_{\theta^1}(\theta^{1\prime\prime}) = f^1(\theta^1) + \nabla_{\theta^1}f^1(\theta^1)^T(\theta^{1\prime\prime} - \theta^1)$$

Applying the proximal operator to the above, and using the results from Section 2.4 (with $x = \theta^{1\prime\prime}$ and $v = \theta^1$), we have:

$$\text{prox}_{\hat{f}^{1(1)}_{\theta^1}}(\theta^1) = \underset{\theta^{1\prime\prime}}{\arg\min}\left(\hat{f}^{1(1)}_{\theta^1}(\theta^{1\prime\prime}) + \frac{1}{2\lambda}||\theta^1 - \theta^{1\prime\prime}||^2_2\right)$$
$$= \theta^1 - \lambda\nabla_{\theta^1}f^1(\theta^1) = \theta^1 - \lambda\nabla_{\theta^1}L^1(\pi_{\theta^1}, \pi_{\theta^2 - \Delta\theta^2(\theta^1)})$$

which recovers the gradient update that LOLA does. $\qquad\square$

**Lemma A.2.** *For any arbitrary fixed $\theta^1$, the naive gradient step in LOLA: $\theta^2 - \Delta\theta^2 = \theta^2 - \lambda\nabla_{\theta^2}L^2(\pi_{\theta^1}, \pi_{\theta^2})$ is equivalent to applying the proximal operator on a first-order approximation (around $\theta^2$) of the objective $L^2(\pi_{\theta^1}, \pi_{\theta^2})$.*

The proof is similar to the one above for Lemma A.1:

*Proof.* Let $\theta^1$ be fixed. Let $f^2(\theta^2) = L^2(\pi_{\theta^1}, \pi_{\theta^2})$.

Apply the first-order Taylor approximation $\hat{f}^{2(1)}_{\theta^2}$ of the objective function (around $\theta^2$), as in the following:

$$\hat{f}^{2(1)}_{\theta^2}(\theta^{2\prime\prime}) = f^2(\theta^2) + \nabla_{\theta^2}f^2(\theta^2)^T(\theta^{2\prime\prime} - \theta^2)$$

Applying the proximal operator to the above, and using the results from Section 2.4 (with $x = \theta^{2\prime\prime}$ and $v = \theta^2$), we have:

$$\text{prox}_{\hat{f}^{2(1)}_{\theta^2}}(\theta^2) = \underset{\theta^{2\prime\prime}}{\arg\min}\left(\hat{f}^{2(1)}_{\theta^2}(\theta^{2\prime\prime}) + \frac{1}{2\lambda}||\theta^2 - \theta^{2\prime\prime}||^2_2\right)$$
$$= \theta^2 - \lambda\nabla_{\theta^2}f^2(\theta^2) = \theta^2 - \lambda\nabla_{\theta^2}L^2(\pi_{\theta^1}, \pi_{\theta^2}) = \theta^2 - \Delta\theta^2$$

thus recovering the gradient update. $\qquad\square$

**Theorem A.3.** *POLA reduces to LOLA when replacing the divergence on policies with the L2 norm on policy parameters and using a first order approximation of both agents' objectives.*

*Proof.* Recall the expression for POLA (Equations 1 and 2):
$$\theta^{1\prime}(\theta^1, \theta^2) = \underset{\theta^{1\prime\prime}}{\arg\min}\left(L^1(\pi_{\theta^{1\prime\prime}}, \pi_{\theta^{2\prime}(\theta^{1\prime\prime}, \theta^2)}) + \beta_{\text{out}}D(\pi_{\theta^1}||\pi_{\theta^{1\prime\prime}})\right)$$
$$\theta^{2\prime}(\theta^{1\prime\prime}, \theta^2) = \underset{\theta^{2\prime\prime}}{\arg\min}\left(L^2(\pi_{\theta^{1\prime\prime}}, \pi_{\theta^{2\prime\prime}}) + \beta_{\text{in}}D(\pi_{\theta^2}||\pi_{\theta^{2\prime\prime}})\right)$$

Replace $L^2(\pi_{\theta^{1\prime\prime}}, \pi_{\theta^{2\prime\prime}})$ with a first order approximation of $L^2(\pi_{\theta^{1\prime\prime}}, \pi_{\theta^2})$ around $\theta^2$, replace $D(\pi_{\theta^2}||\pi_{\theta^{2\prime\prime}})$ with $||\theta^2 - \theta^{2\prime\prime}||^2_2$, and set $\beta_{\text{in}} = \frac{1}{2\lambda}$. Then by Lemma A.2, $\theta^{2\prime}(\theta^{1\prime\prime}, \theta^2) = \theta^2 - \Delta\theta^2(\theta^{1\prime\prime})$, resulting in the following equation (which is Outer POLA in Section 3.3):

$$\theta^{1\prime}(\theta^1, \theta^2) = \underset{\theta^{1\prime\prime}}{\arg\min}\left(L^1(\pi_{\theta^{1\prime\prime}}, \pi_{\theta^2 - \Delta\theta^2(\theta^{1\prime\prime})}) + \beta_{\text{out}}D(\pi_{\theta^1}||\pi_{\theta^{1\prime\prime}})\right)$$

Replace $D(\pi_{\theta^1}||\pi_{\theta^{1\prime\prime}})$ with $||\theta^1 - \theta^{1\prime\prime}||^2_2$, set $\beta_{\text{out}} = \frac{1}{2\lambda}$, and replace $L^1(\pi_{\theta^{1\prime\prime}}, \pi_{\theta^2 - \Delta\theta^2(\theta^{1\prime\prime})})$ with the first order approximation of $L^1(\pi_{\theta^1}, \pi_{\theta^2})$ around $\theta^1$, $\hat{f}^{1(1)}_{\theta^1}(\theta^{1\prime\prime})$:

$$\theta^{1\prime}(\theta^1, \theta^2) = \underset{\theta^{1\prime\prime}}{\arg\min}\left(\hat{f}^{1(1)}_{\theta^1}(\theta^{1\prime\prime}) + \frac{1}{2\lambda}||\theta^1 - \theta^{1\prime\prime}||^2_2\right)$$

This is exactly $\text{prox}_{\hat{f}^{1(1)}_{\theta^1}}(\theta^1)$, so by Lemma A.1, this reduces to LOLA. $\qquad\square$

---

[2]Note that this is different from the first-order approximation used in the original LOLA paper, as that approximation is a Taylor series expansion of $L^1(\pi_{\theta^1}, \pi_{\theta^2})$, whereas here we use a Taylor series expansion of $L^1(\pi_{\theta^1}, \pi_{\theta^2 - \Delta\theta^2(\theta^1)})$

## A.4 LOLA-DiCE

DiCE [Foerster et al., 2018b] introduces an infinitely differentiable estimator for unbiased higher-order Monte Carlo gradient estimates. DiCE introduces a new operator $\boxdot$ which operates on a set of stochastic nodes $\mathcal{W}$, where:

$$\boxdot(\mathcal{W}) = \exp(\tau - \bot(\tau)) \text{ and } \tau = \sum_{w \in \mathcal{W}} \log p(w; \boldsymbol{\theta}). \tag{4}$$

$\bot$ is a stop-gradient operator (detach in Pytorch). The loss for agent 1, on a single rollout using policies $(\pi_{\theta^1}, \pi_{\theta^{2\prime}})$, with LOLA-DiCE is:

$$\mathcal{L}^1_{\boxdot(\pi_{\theta^1}, \pi_{\theta^{2\prime}})} = -\sum_{t=0}^{T} \boxdot(a_{\leq t}) \gamma^t r_t^1 \tag{5}$$

where $a_{\leq t}$ denotes the set of actions all agents took at time step $t$ or earlier. Algorithm 4 provides pseudo-code for LOLA-DiCE.

---
**Algorithm 4** LOLA-DiCE 2-agent formulation: update for agent 1
---
    **Input:** Policy parameters $\boldsymbol{\theta} = \{\theta^1, \theta^2\}$, learning rates $\alpha, \eta$, number of inner steps $K$
    Initialize: $\boldsymbol{\theta}' \leftarrow \boldsymbol{\theta}$
    **for** $k$ in $1...K$ **do**
        Rollout trajectories using policies $(\pi_{\theta^1}, \pi_{\theta^{2\prime}})$
        $\theta^{2\prime} \leftarrow \theta^{2\prime} - \eta \nabla_{\theta^{2\prime}} \mathcal{L}^2_{\boxdot(\pi_{\theta^1}, \pi_{\theta^{2\prime}})}$
    **end for**
    Rollout trajectories using policies $(\pi_{\theta^1}, \pi_{\theta^{2\prime}})$
    $\theta^{1\prime} \leftarrow \theta^1 - \alpha \nabla_{\theta^1} \mathcal{L}^1_{\boxdot(\pi_{\theta^1}, \pi_{\theta^{2\prime}})}$
    **Output:** $\theta^{1\prime}$
---

LOLA-DiCE [Foerster et al., 2018b] replicated the original LOLA policy gradient results [Foerster et al., 2018a] in a way that was more direct, efficient, and stable, supported by experiments with tabular policies in the one-step memory IPD.

## A.5 Loaded DiCE

*Loaded DiCE* [Farquhar et al., 2019] rewrites the DiCE objective (5) as:

$$\mathcal{L}^1_{\boxdot(\pi_{\theta^1}, \pi_{\theta^{2\prime}})} = -\sum_{t=0}^{T} \gamma^t \left( \boxdot(a_{\leq t}) - \boxdot(a_{<t}) \right) \sum_{l=t}^{T} \gamma^{l-t} r_l^1 \tag{6}$$

where $a_{<t}$ denotes the set of actions all agents took before time step $t$.

Farquhar et al. [2019] showed Equation 6 has the same gradients as Equation 5. $\sum_{l=t}^{T} \gamma^{l-t} r_l^1$ in Equation 6 is then replaced with an advantage function: $A^1(s_t, a_t)$. Thus, *loaded DiCE* incorporates a baseline for variance reduction with DiCE.

In all our experiments with rollouts (Sections 4.2 and 4.3), we use *loaded DiCE* (Equation 6) with generalized advantage estimation (see Appendix A.6 and Schulman et al. [2015b]) for the baseline.

## A.6 Generalized Advantage Estimation

Generalized advantage estimation [Schulman et al., 2015b], introduces the following advantage estimator for time step $t$:

$$\hat{A}_t^{GAE(\gamma, \lambda)} \triangleq (1 - \lambda)(\hat{A}_t^{(1)} + \lambda \hat{A}_t^{(2)} + \lambda^2 \hat{A}_t^{(3)} + ...)$$

$$\hat{A}_t^{(k)} \triangleq \sum_{l=0}^{k-1} \gamma^l \delta_{t+l}^V = -V(s_t) + r_t + \gamma r_{t+1} + ... + \gamma^{k-1} r_{t+k-1} + \gamma^k V(s_{t+k})$$

When $\lambda = 0$, the advantage estimator is the one-step Bellman residual, whereas $\lambda = 1$ is equivalent to Monte Carlo estimation (in the finite horizon setting, it extrapolates the Monte Carlo estimation with the final state value). In our finite-horizon experiments, we use a finite sum:

$$\hat{A}_t^{GAE(\gamma,\lambda,T)} \triangleq (1 - \lambda) \sum_{t'=1}^{T} \lambda^{t'-1} \hat{A}_t^{(t')}$$

which is the same as what *loaded DiCE* does [Farquhar et al., 2019].

When the value function $V$ is not completely accurate, the advantage estimator is biased. Using a lower value of $\lambda$ increases bias, but reduces variance.

### A.7   Training the Critic

Advantage estimation requires a value function (critic) $V$. To train the critic, we minimize the mean squared error of the Monte Carlo return extended by the value in the final state:

$$\mathcal{L}_{critic} = \sum_{t=0}^{T} ([-V(s_t)] + [r_t + \gamma r_{t+1} + ... + \gamma^{T-t-1} r_{T-1} + \gamma^{T-t} V(s_T)])^2$$

averaged over samples in the batch.

### A.8   POLA-DiCE with Repeated Training on the Same Samples

POLA-DiCE can require a lot of environment rollouts. One idea for improving sample efficiency and reducing training time is developing a modification like Schulman et al. [2017] or Kakade and Langford [2002] that allows for repeated training on the same set of samples.

We define a new operator similar to (4) from LOLA-DiCE:

$$\boxdot(\mathcal{W}) = \exp(\tau' - \bot(\tau')) \text{ and } \tau' = \sum_{w \in \mathcal{W}} p(w; \boldsymbol{\theta})/p(w; \boldsymbol{\theta}_{old}) \tag{7}$$

where $\boldsymbol{\theta}_{old}$ denotes policy parameters used on the first rollout (step $k = 1$ for the inner loop and step $m = 1$ for the outer loop).

Our new loss for agent 1 is:

$$\mathcal{L}^1_{\boxdot(\pi_{\theta 1}, \pi_{\theta 2'})} = \sum_{t=0}^{T} (\boxdot(a_{\leq t}) - \boxdot(a_{<t})) \gamma^t A^1(s_t, a_t)$$

This is the *loaded DiCE* formulation (6), but with probability ratios instead of log probabilities, which lets us make multiple updates on the same batch of rollouts. We provide pseudo-code in Algorithm 5; we call this POLA-DiCE-PPO.

On the first update step ($m = 1$ and $k = 1$), $\theta_{old} = \bot(\theta)$, so $\nabla_{\theta 2''} \mathcal{L}^2_{\boxdot(\pi_{\theta 1''}, \pi_{\theta 2''})} = \nabla_{\theta 2''} \mathcal{L}^2_{\boxdot(\pi_{\theta 1''}, \pi_{\theta 2''})}$ and $\nabla_{\theta 1''} \mathcal{L}^1_{\boxdot(\pi_{\theta 1''}, \pi_{\theta 2''})} = \nabla_{\theta 1''} \mathcal{L}^1_{\boxdot(\pi_{\theta 1''}, \pi_{\theta 2''})}$. Thus, when inner steps $K = 1$, outer steps $M = 1$, and $\beta_{in}, \beta_{out} = 0$, POLA-DiCE-PPO is equivalent to LOLA-DiCE.

Algorithm 5 repeats training only on the inner loop. A similar repeated training procedure can be used on the outer loop, for even greater sample efficiency. However, this works only if the higher order gradients calculated from the initial rollout remain accurate as the policy changes.

We present this section only in the Appendix because we empirically found a larger number of outer steps to be more important than a larger number of inner steps, and that repeatedly training on the same samples for the outer loop resulted in learning reciprocity-based cooperation significantly less consistently. In future work, modifications such as periodic rollouts every $x < M$ outer steps may provide improved sample efficiency with less deterioration in performance.

This objective can also be clipped, like the clipped version of PPO [Schulman et al., 2017]; future work could explore this and compare to the KL penalty version.

---
**Algorithm 5** POLA-DiCE-PPO 2-agent formulation: update for agent 1
---
**Input:** Policy parameters $\theta^1, \theta^2$, learning rates $\alpha_1, \alpha_2$, penalty hyperparameters $\beta_{\text{in}}, \beta_{\text{out}}$, number of outer steps $M$ and inner steps $K$
Initialize: $\theta^{1\prime\prime} \leftarrow \theta^1$
**for** $m$ in $1...M$ **do**
    Initialize: $\theta^{2\prime\prime} \leftarrow \theta^2$
    **for** $k$ in $1...K$ **do**
        **if** k = 1 **then**
            Rollout trajectories using policies $(\pi_{\theta^{1\prime\prime}}, \pi_{\theta^{2\prime\prime}})$. Save states $s^{\text{in}}_{\leq T}$ from the trajectories
        **end if**
        $\theta^{2\prime\prime} \leftarrow \theta^{2\prime\prime} - \alpha_2 \nabla_{\theta^{2\prime\prime}} (\mathcal{L}^2_{\boxdot(\pi_{\theta^{1\prime\prime}}, \pi_{\theta^{2\prime\prime}})} + \beta_{\text{in}} D(\pi_{\theta^2}, \pi_{\theta^{2\prime\prime}} | s^{\text{in}}_{\leq T}))$
    **end for**
    Rollout trajectories with states $s^{\text{out}}_{\leq T}$ using policies $(\pi_{\theta^{1\prime\prime}}, \pi_{\theta^{2\prime\prime}})$
    $\theta^{1\prime\prime} \leftarrow \theta^{1\prime\prime} - \alpha_1 \nabla_{\theta^{1\prime\prime}} (\mathcal{L}^1_{\boxdot(\pi_{\theta^{1\prime\prime}}, \pi_{\theta^{2\prime\prime}})} + \beta_{\text{out}} D(\pi_{\theta^1}, \pi_{\theta^{1\prime\prime}} | s^{\text{out}}_{\leq T}))$
**end for**
**Output:** $\theta^{1\prime\prime}$
---

## A.9 POLA N-Agent Formulation

Consider again agent 1's perspective. Agent 1 solves for the following policy:

$$\theta^{1\prime}(\theta^1, \theta^2, ..., \theta^N) = \underset{\theta^{1\prime\prime}}{\arg\min} \left( L^1(\pi_{\theta^{1\prime\prime}}, \pi_{\theta^{2\prime}(\theta^{1\prime\prime}, \theta^2, ..., \theta^N)}, ..., \pi_{\theta^{N\prime}(\theta^{1\prime\prime}, \theta^2, ..., \theta^N)}) + \beta_{\text{out}} D(\pi_{\theta^1} || \pi_{\theta^{1\prime\prime}}) \right)$$
(8)

where $D(\pi_{\theta^i} || \pi_{\theta^{i\prime\prime}})$ is again shorthand for a general divergence defined on policies. For $\theta^{2\prime}, ..., \theta^{N\prime}$ in Equation 8, we choose the following proximal updates:

$$\theta^{2\prime}(\theta^{1\prime\prime}, \theta^2, ..., \theta^N) = \underset{\theta^{2\prime\prime}}{\arg\min} \left( L^2(\pi_{\theta^{1\prime\prime}}, \pi_{\theta^{2\prime\prime}}, \pi_{\theta^3}..., \pi_{\theta^N}) + \beta_{\text{in}} D(\pi_{\theta^2} || \pi_{\theta^{2\prime\prime}}) \right)$$

$$\vdots$$

$$\theta^{N\prime}(\theta^{1\prime\prime}, \theta^2, ..., \theta^N) = \underset{\theta^{N\prime\prime}}{\arg\min} \left( L^N(\pi_{\theta^{1\prime\prime}}, \pi_{\theta^2}, ..., \pi_{\theta^{N-1}}, \pi_{\theta^{N\prime\prime}}) + \beta_{\text{in}} D(\pi_{\theta^N} || \pi_{\theta^{N\prime\prime}}) \right)$$

In short, agent 1 assumes all other agents $i$ find the argmin of their loss, assuming the policies of other agents $j \neq i$ are fixed (using agent 1's updated policy $\theta^{1\prime\prime}$ and the original policies of all other agents).

## A.10 POLA-DiCE N-Agent Formulation

Let $\pi_{\theta^{-1, -i}} \triangleq \{\pi_{\theta^2}, ..., \pi_{\theta^N}\} \setminus \pi_{\theta^i}$ be shorthand for all of the policies except those of agent 1 and agent $i$. Algorithm 6 provides an N-agent formulation of POLA-DiCE. The idea is similar to Appendix A.9; in the inner loop, each agent updates assuming other agents' policies are fixed.

# B Appendix: Experiment and Hyperparameter Details

## B.1 One-Step Memory IPD

### B.1.1 Exact Loss Calculation

With one step of memory, we can directly build the transition probability matrix $\mathcal{P}$ and starting state distribution $\mathbf{p}_0$ given all agents' policies. Foerster et al. [2018a] in their Appendix A.2 derived the exact loss: $L^i = -\mathbf{p}_0^T (I - \gamma \mathcal{P})^{-1} R^i$. This can then be directly used for gradient updates.

### B.1.2 Example Showing how the Cooperation Factor Recovers the IPD Reward Structure

Below we show that $f = 4/3$ recovers the IPD from Foerster et al. [2018a]:

---

**Algorithm 6** POLA-DiCE N-agent formulation: update for agent 1

---

**Input:** Policy parameters $\theta^1, \theta^2, ..., \theta^N$, learning rates $\alpha_1, \alpha_2, ..., \alpha_N$, penalty hyperparameters $\beta_{\text{in}}^2, ..., \beta_{\text{in}}^N, \beta_{\text{out}}$, number of outer steps $M$ and inner steps $K$
Initialize: $\theta^{1\prime\prime} \leftarrow \theta^1$
**for** $m$ in $1...M$ **do**
    **for** $i$ in $2...N$ **do**
        Initialize: $\theta^{i\prime\prime} \leftarrow \theta^i$
        **for** $k$ in $1...K$ **do**
            Rollout trajectories with states $s_{\leq T}^{\text{in}}$ using policies $(\pi_{\theta^{1\prime\prime}}, \pi_{\theta^{i\prime\prime}}, \pi_{\theta-1,-i})$
            $\theta^{i\prime\prime} \leftarrow \theta^{i\prime\prime} - \alpha_i \nabla_{\theta^{i\prime\prime}}(\mathcal{L}^i_{\odot(\pi_{\theta^{1\prime\prime}}, \pi_{\theta^{i\prime\prime}}, \pi_{\theta-1,-i})} + \beta_{\text{in}}^i D(\pi_{\theta^i}, \pi_{\theta^{i\prime\prime}}|s_{\leq T}^{\text{in}}))$
        **end for**
    **end for**
    Rollout trajectories with states $s_{\leq T}^{\text{out}}$ using policies $(\pi_{\theta^{1\prime\prime}}, \pi_{\theta^{2\prime\prime}}, ..., \pi_{\theta^{N\prime\prime}})$
    $\theta^{1\prime\prime} \leftarrow \theta^{1\prime\prime} - \alpha_1 \nabla_{\theta^{1\prime\prime}}(\mathcal{L}^1_{\odot(\pi_{\theta^{1\prime\prime}}, \pi_{\theta^{2\prime\prime}}, ..., \pi_{\theta^{N\prime\prime}})} + \beta_{\text{out}} D(\pi_{\theta^1}, \pi_{\theta^{1\prime\prime}}|s_{\leq T}^{\text{out}}))$
**end for**
**Output:** $\theta^{1\prime\prime}$

---

Let $M$ be the number of agents who cooperate, $D(M)$ be the payoff to each defector, and $C(M)$ be the payoff to each cooperator. Then $C(2) = 1/3$, $C(1) = -1/3$, $D(1) = 2/3$, $D(0) = 0$, and we have the payoff matrix:

| P1/P2 | C | D |
|-------|---|---|
| C | (1/3, 1/3) | (-1/3, 2/3) |
| D | (2/3, -1/3) | (0, 0) |

Multiplying all rewards by 3 (equivalent to scaling the learning rate), subtracting 2 from all rewards (which preserves the ordering of policies by expected reward, and is equivalent under an accurately learned value function baseline), then gives the payoff matrix:

| P1/P2 | C | D |
|-------|---|---|
| C | (-1, -1) | (-3, 0) |
| D | (0, -3) | (-2, -2) |

which is the 2-player IPD with the reward structure given in Foerster et al. [2018a].

### B.1.3 Function Approximation Setup

With function approximation, our state representation is a one-hot vector with 3 dimensions (defect, cooperate, and start state) for each agent's past action. Thus, with two agents, each agent's input is two 3-d one-hot vectors, which we flatten to a single 6-d vector. We use this representation as it has size $\Theta(N)$, whereas a single one-hot vector over all possible combinations of actions has size $\Theta(2^N)$; our representation is conducive to future experiments with large $N$.

### B.1.4 Hyperparameter Settings

Typical neural network weight initializations (e.g. Gaussian) produce policies that are close to random (cooperating with probability close to 0.5 in each state). Our policy probability initializations are close to random throughout; this helps provide comparable results between tabular and neural network policies. Naive learning can find TFT if initialized sufficiently close to it, but never finds TFT with policies initialized close to random.

Empirically, for policies initialized close to random and for sufficiently large $\eta$, LOLA always updates the policy toward cooperating *iff* the opponent last cooperated. Thus, with sufficiently large learning rate $\alpha$, LOLA finds TFT in the tabular setting (for $f > 1$). However, LOLA no longer finds TFT with large learning rates when the policy parameterization is a neural network function approximator or a transformed tabular policy (Figure 3).

For Figure 3, we show results from the best set of hyperparameters (highest % TFT found). We tuned inner and outer learning rates $(\eta, \alpha)$ for LOLA using a greedy heuristic; we increased or decreased

Table 1: Average Policies (Probability of Cooperation by State) Learned in IPD for Various Policy Parameterizations

| Algorithm | Parameterization (Self & Opponent) | DD | DC | CD | CC | Start |
|---|---|---|---|---|---|---|
| Contribution Factor 1.1 | | | | | | |
| Naive Learning | Tabular | 0.00 | 0.06 | 0.06 | 0.16 | 0.01 |
| LOLA | Tabular | 0.00 | 1.00 | 0.00 | 1.00 | 1.00 |
| LOLA | Neural net | 0.00 | 0.01 | 0.00 | 0.03 | 0.02 |
| LOLA | Pre-condition | 0.00 | 0.00 | 0.97 | 0.00 | 0.00 |
| Outer POLA | Tabular | 0.00 | 0.97 | 0.45 | 1.00 | 0.94 |
| Outer POLA | Neural net | 0.00 | 0.94 | 0.49 | 1.00 | 0.94 |
| Outer POLA | Pre-condition | 0.01 | 0.29 | 0.26 | 0.87 | 0.60 |
| Contribution Factor 1.25 | | | | | | |
| Naive Learning | Tabular | 0.00 | 0.07 | 0.07 | 0.18 | 0.02 |
| LOLA | Tabular | 0.00 | 1.00 | 0.00 | 1.00 | 1.00 |
| LOLA | Neural net | 0.00 | 0.02 | 0.01 | 0.06 | 0.02 |
| LOLA | Pre-condition | 0.00 | 0.00 | 0.96 | 0.00 | 0.00 |
| Outer POLA | Tabular | 0.08 | 0.97 | 0.18 | 1.00 | 0.95 |
| Outer POLA | Neural net | 0.01 | 0.87 | 0.47 | 1.00 | 0.85 |
| Outer POLA | Pre-condition | 0.12 | 0.97 | 0.23 | 1.00 | 0.77 |
| Contribution Factor 1.33 | | | | | | |
| Naive Learning | Tabular | 0.00 | 0.07 | 0.07 | 0.19 | 0.02 |
| LOLA | Tabular | 0.00 | 1.00 | 0.00 | 1.00 | 1.00 |
| LOLA | Neural net | 0.03 | 0.35 | 0.06 | 0.41 | 0.15 |
| LOLA | Pre-condition | 0.00 | 0.00 | 0.96 | 0.00 | 0.00 |
| Outer POLA | Tabular | 0.13 | 0.96 | 0.08 | 1.00 | 0.94 |
| Outer POLA | Neural net | 0.02 | 0.85 | 0.45 | 0.99 | 0.68 |
| Outer POLA | Pre-condition | 0.18 | 0.99 | 0.30 | 1.00 | 0.76 |
| Contribution Factor 1.4 | | | | | | |
| Naive Learning | Tabular | 0.00 | 0.07 | 0.07 | 0.20 | 0.02 |
| LOLA | Tabular | 0.00 | 1.00 | 0.00 | 1.00 | 1.00 |
| LOLA | Neural net | 0.25 | 0.87 | 0.37 | 0.88 | 0.65 |
| LOLA | Pre-condition | 0.00 | 0.00 | 0.96 | 0.00 | 0.00 |
| Outer POLA | Tabular | 0.13 | 0.96 | 0.12 | 1.00 | 0.95 |
| Outer POLA | Neural net | 0.02 | 0.87 | 0.44 | 0.99 | 0.76 |
| Outer POLA | Pre-condition | 0.21 | 0.98 | 0.33 | 1.00 | 0.75 |
| Contribution Factor 1.6 | | | | | | |
| Naive Learning | Tabular | 0.00 | 0.09 | 0.09 | 0.23 | 0.05 |
| LOLA | Tabular | 0.00 | 1.00 | 0.00 | 1.00 | 1.00 |
| LOLA | Neural net | 0.15 | 0.98 | 0.41 | 0.97 | 0.67 |
| LOLA | Pre-condition | 0.10 | 0.10 | 0.87 | 0.10 | 0.10 |
| Outer POLA | Tabular | 0.05 | 0.90 | 0.07 | 1.00 | 0.91 |
| Outer POLA | Neural net | 0.03 | 0.95 | 0.28 | 0.99 | 0.87 |
| Outer POLA | Pre-condition | 0.08 | 0.98 | 0.20 | 1.00 | 1.00 |

values until we no longer got better results. For *outer POLA*, we tuned $\eta$ and $\beta_{\text{out}}$, and did less tuning on the outer learning rate, which matters only for convergence speed and stability. For the exact set of hyperparameters, see: `https://github.com/Silent-Zebra/POLA`.

### B.1.5 Additional Detailed IPD Results

In Table 1, we show the average probability of cooperation in each state of the IPD for each of the algorithms and contribution factors shown in Figure 3. Here, DD denotes both agents last defected,

Table 2: Average Policies (Probability of Cooperation by State) Learned in IPD for Various Opponent Model Policy Parameterizations

| Algorithm | Parameterization (Opponent Model) | DD | DC | CD | CC | Start |
|-----------|-----------------------------------|------|------|------|------|-------|
| Contribution Factor 1.33 | | | | | | |
| LOLA | Tabular | 0.02 | 0.99 | 0.17 | 1.00 | 0.93 |
| LOLA | Neural net | 0.00 | 0.99 | 0.03 | 1.00 | 0.97 |
| LOLA | Pre-condition | 0.00 | 0.10 | 0.07 | 0.23 | 0.08 |
| POLA | Tabular | 0.01 | 0.97 | 0.05 | 1.00 | 0.97 |
| POLA | Neural net | 0.01 | 0.97 | 0.05 | 1.00 | 0.97 |
| POLA | Pre-condition | 0.12 | 0.94 | 0.02 | 1.00 | 0.85 |

DC denotes the agent last defected while the opponent last cooperated, CD denotes the agent last cooperated while the opponent last defected, CC denotes both agents last cooperated, and Start is the starting state. Numbers are averaged over 20 runs and averaged over both agents. *Outer POLA* learns reciprocity-based cooperation across contribution factor and policy parameterization settings much more consistently than LOLA.

### B.1.6 IPD Experiments with Varying Opponent Model Parameterizations

We revisit the IPD with opponent modeling, considering agents with tabular policies but varying parameterizations of opponent models: tabular, neural network function approximation, and pre-conditioned tabular models. These parameterizations are the same as described previously except for the pre-conditioned tabular model, for which we now use:

$$\mathbf{Q}^1 = \begin{pmatrix} 1 & 0 & 0 & 0 & 0 \\ -2 & 1 & 0 & 0 & 0 \\ -2 & 0 & 1 & 0 & 0 \\ -2 & 0 & 0 & 1 & 0 \\ -2 & 0 & 0 & 0 & 1 \end{pmatrix}, \mathbf{Q}^2 = \begin{pmatrix} 1 & 0 & 0 & 0 & 0 \\ -2 & 1 & 0 & 0 & 0 \\ -2 & 0 & 1 & 0 & 0 \\ -2 & 0 & 0 & 1 & 0 \\ -2 & 0 & 0 & 0 & 1 \end{pmatrix}$$

This set of transformations again only changes basis and lets us illustrate the difference between LOLA and POLA. We show results in Table 2. LOLA fails to learn reciprocity-based cooperation for the pre-conditioned tabular opponent model, whereas POLA learns reciprocity-based cooperation across all parameterizations.

LOLA learns reciprocity-based cooperation well even with a neural net opponent policy. We believe this is because the inner player policy update (usually towards defecting in each state) is less sensitive to policy parameterization than the outer player policy update which requires second order gradients and learning reciprocity.

The version of POLA we use here is similar to POLA-DiCE (Algorithm 2) except with exact losses and the uniform distribution KL penalty; we provide pseudocode in Algorithm 7.

---

**Algorithm 7** POLA direct approximation 2-agent formulation: update for agent 1

---

**Input:** Policy parameters $\theta^1, \theta^2$, learning rates $\alpha_1, \alpha_2$, penalty hyperparameters $\beta_{\text{in}}, \beta_{\text{out}}$, number of outer steps $M$ and inner steps $K$
Initialize: $\theta^{1\prime\prime} \leftarrow \theta^1$
**for** $m$ in $1...M$ **do**
   Initialize: $\theta^{2\prime\prime} \leftarrow \theta^2$
   **for** $k$ in $1...K$ **do**
     $\theta^{2\prime\prime} \leftarrow \theta^{2\prime\prime} - \alpha_2 \nabla_{\theta^{2\prime\prime}}(L^2(\pi_{\theta^{1\prime\prime}}, \pi_{\theta^{2\prime\prime}}) + \beta_{\text{in}}(\mathbb{E}_{s \sim U(\mathcal{S})}[D_{KL}(\pi_{\theta^2}(s)||\pi_{\theta^{2\prime\prime}}(s))]))$
   **end for**
   $\theta^{1\prime\prime} \leftarrow \theta^{1\prime\prime} - \alpha_1 \nabla_{\theta^{1\prime\prime}}(L^1(\pi_{\theta^{1\prime\prime}}, \pi_{\theta^{2\prime\prime}}) + \beta_{\text{out}}(\mathbb{E}_{s \sim U(\mathcal{S})}[D_{KL}(\pi_{\theta^1}(s)||\pi_{\theta^{1\prime\prime}}(s))]))$
**end for**
**Output:** $\theta^{1\prime\prime}$

---

We use 100-200 inner steps for POLA and 1 outer step; since we are only changing the opponent model parameterization, invariance on the inner loop is most important. This is a fairly small number of steps which does not achieve full invariance, causing results to vary slightly across parameterizations. We

assume unlimited batch size for learning the opponent model, which is equivalent to directly learning from the policy, to separate the effects of noise from opponent modeling. Detailed hyperparameters are available in the codebase (`https://github.com/Silent-Zebra/POLA`).

## B.2   IPD Full History Details

We use the state representation as discussed in Appendix B.1.3. Agents condition actions on the entire state history up to the current time step. Policies are parameterized by a fully connected input layer with 64 hidden units, a ReLU nonlinearity, a GRU cell, and a fully connected output layer. We use a batch size of 2000 (parallel environment rollouts), discount rate $\gamma = 0.96$, and rollout for $T = 50$ steps in the environment. For both LOLA-DiCE and POLA-DiCE, we use a simple gradient step on the inner loop and the Adam optimizer with default betas on the outer loop, as LOLA-DiCE [Foerster et al., 2018b] does.

For all algorithms, we use *loaded DiCE* (Appendix A.5) with GAE (Appendix A.6) with $\lambda = 1$, an outer learning rate of 0.003, and a learning rate for the critic (value function) of 0.0005. For LOLA-DiCE, we use an inner learning rate of 0.05, but we also tried the values: [0.005, 0.015, 0.02, 0.03, 0.07, 0.1, 0.2] and got similar or worse results. We also tried a few settings with lower and higher outer learning rates and value function learning rates, and 2 inner steps; none learned reciprocity-based cooperation more consistently.

For POLA-DiCE, we use 2 inner steps and 200 outer steps, with $\beta_{\text{in}} = 10$ and $\beta_{\text{out}} = 100$, and an inner learning rate of 0.005. Results are not very sensitive to $\beta_{\text{out}}$; we got similar results with $\beta_{\text{out}} = 30$ and $\beta_{\text{out}} = 200$. We did not extensively tune hyperparameters for POLA-DiCE in this setting, so it is likely that similar results can be reproduced with fewer outer steps. Results are more sensitive to the inner learning rate and $\beta_{\text{in}}$, but we expect using more inner steps would lessen this sensitivity. One challenge with taking more inner steps is the memory requirement, which forces a tradeoff with batch size. We update the critic after each policy update on both the inner and outer loop (for the corresponding agent or opponent model).

To learn the opponent model in POLA-OM, we use 200 environment rollouts (of 2000 batch size each) between each set of POLA updates. The opponent model architecture is the same as the agent's. We learn the opponent's value function in the same way as the agent's own value function, but with the reward and states of the other agent. We use learning rates of 0.005 for the policy model and 0.0005 for the value model. For other hyperparameters, we use the same settings for POLA-DiCE with and without opponent modeling.

In Figure 4 we choose the number of outer steps as the x-axis because the outer steps are policy updates that are actually made; inner step updates are not saved, and are used only in the gradient calculation of outer step updates. Each inner step currently requires an environment rollout, though this can be mitigated in future work (e.g. Appendix A.8), another reason why we consider outer steps more representative of sample efficiency. Strictly comparing environment rollouts would horizontally stretch the lines for POLA-DiCE relative to LOLA-DiCE by a factor of 1.5 (3 environment rollouts per outer step for POLA-DiCE vs. 2 for LOLA-DiCE); this would not change the conclusions drawn in the paper.

## B.3   Coin Game Details

Our environment implementation adheres to Figure 3 in Foerster et al. [2018a] where two agents that step on the same coin at the same time both collect the coin. In previous experiments, splitting the coin 50-50 between agents gave very similar results.

The LOLA results in Figure 5 cannot be directly compared with those in Foerster et al. [2018a] for several reasons. Our implementation of the coin game environment fixes bugs such as ties always being broken in favour of the red agent (see: https://github.com/alshedivat/lola/issues/9), which make the original results irreproducible. We use LOLA-DiCE instead of the original LOLA-PG formulation. We rollout for fewer steps to reduce computation time and memory requirements. Our policy parameterization has more hidden units and uses a GRU, which should make it more expressive than the originally used RNN, and may also make optimization more difficult.

Same as the IPD with full history, our GRUs use 64 hidden units with a fully connected input layer with ReLU nonlinearity and a linear output layer. We use a batch size of 2000 (parallel environment

rollouts), discount rate $\gamma = 0.96$, and rollout for $T = 50$ steps in the environment. For both LOLA-DiCE and POLA-DiCE, we use a simple gradient step on the inner loop and the Adam optimizer with default betas on the outer loop, as LOLA-DiCE [Foerster et al., 2018b] does.

For all algorithms, we use *loaded DiCE* (Appendix A.5) with GAE (Appendix A.6) with $\lambda = 1$, an outer learning rate of 0.003, and a learning rate for the critic (value function) of 0.0005. For LOLA-DiCE, we use an inner learning rate of 0.003, but we also tried the values: [0.001, 0.005, 0.01, 0.02, 0.05, 0.1, 0.2] and got similar results. We also tried a few settings with lower and higher outer learning rates and value function learning rates, and 2 inner steps; none learned reciprocity-based cooperation more consistently.

For POLA-DiCE, we use 2 inner steps and 200 outer steps, with $\beta_{in} = 5$ and $\beta_{out} = 150$, and an inner learning rate of 0.02. We update the critic after each policy update on both the inner and outer loop (for the corresponding agent or opponent model).

To learn the opponent model in POLA-OM, we use 200 environment rollouts (of 2000 batch size each) between each set of POLA updates. The opponent model architecture is the same as the agent's. We learn the opponent's value function in the same way as the agent's own value function, but with the reward and states of the other agent. We use learning rates of 0.005 for the policy model and 0.0005 for the value model. For POLA-OM, we use 4 inner steps with $\beta_{in} = 10$ and an inner learning rate of 0.01; other hyperparameters are the same as POLA-DiCE without opponent modeling. The additional inner steps and higher $\beta_{in}$ provide greater invariance to the opponent model, which helps in the coin game. We tried a few settings with even more inner steps but those did not learn reciprocity-based cooperation more consistently. We believe this may be due to memory constraints forcing smaller batch sizes (and thus more noise from environment rollouts) with more inner steps.

In Figure 5 we choose the number of outer steps as the x-axis because the outer steps are policy updates that are actually made; inner step updates are not saved, and are used only in the gradient calculation of outer step updates. Each inner step currently requires an environment rollout, though this can be mitigated in future work (e.g. Appendix A.8), another reason why we consider outer steps more representative of sample efficiency. Strictly comparing environment rollouts would horizontally stretch the lines for POLA-DiCE relative to LOLA-DiCE by a factor of 1.5 (3 environment rollouts per outer step for POLA-DiCE vs. 2 for LOLA-DiCE); this would not change the conclusions drawn in the paper.

### B.4 Code Details

Parts of code were adapted from `https://github.com/alexis-jacq/LOLA_DiCE` [Foerster et al., 2018b] and `https://github.com/aletcher/stable-opponent-shaping` [Letcher et al., 2018]. Both use the MIT license, which grants permission free of charge for subsequent use, modification, and distribution.

### B.5 Compute Usage

For the IPD with one-step memory (Section 4.1), experiments were run on CPUs provided free of charge by Google Colaboratory. Most experiments required only a small amount of compute (taking minutes to run).

For the IPD with full history (Section 4.2) and the coin game (Section 4.3), experiments were run on GPUs on an internal cluster. GPUs were either NVIDIA Tesla T4 or NVIDIA Tesla P100. Coin game experiments took around 1 full day (24 hours) to run for 1 seed on 1 GPU, whereas the IPD with full history experiments took around 8-10 hours for each seed.

The total amount of compute used, including during the experimentation phase, was significantly higher than that used for Figures 4 and 5.

## C  Societal Impact

We do not anticipate any immediate societal impact from this work; at the time of writing, there is no direct real world application. That said, we hope this work helps produce socially benefi-cial outcomes when autonomous learning agents interact, which is critical for future real-world

deployment. However, while opponent shaping helps in the social dilemma settings we tested, it could cause undesirable consequences in other settings. For example, pricing algorithms learning reciprocity-based cooperation would be tantamount to collusion. In such cases, POLA could learn undesirable behaviour in a way that is invariant to policy parameterization. Overall though, we expect the potential positive impact of our work to outweigh the potential negative impact.