# OpenReview forum: "Proximal Learning With Opponent-Learning Awareness"
_NeurIPS.cc/2022/Conference — NeurIPS 2022 Accept_

### Official Review · Reviewer_kxwY · 2022-06-29

**Rating:** 7
**Confidence:** 5
**Soundness:** 4 excellent
**Presentation:** 4 excellent
**Contribution:** 3 good

**Summary:**

The paper proposes POLA, an extension to LOLA. LOLA is an opponent-shaping technique that enables RL agents to converge to conditional cooperation in mixed environments. A downside of LOLA is that it assumes specific agent parameterization and its updates depend on this parameterization. Authors empirically demonstrate that different parameterization can result in different equilibrium. POLA mitigates this downside by shifting the opponent shaping from the parameter space to the policy space. Empirically, POLA outperforms LOLA (in frequency of converging to tit-for-tat) in iterated prisoner dilemma with history, as well as in coin game (a temporally-extended social dilemma).

**Questions:**

Sec 3.1: is there an explanation of why these Q^i are used? Why does LOLA fail (or why does sensitivity to parameterezation results in different equilibrium)? Does LOLA fail on other Q^i that only change basis?

**Limitations:**

Limitations and societal impact are adequately addressed.

**Strengths And Weaknesses:**

Overall, the paper presents a clear and solid improvement of a relevant algorithm. This impovement is a bit marginal, and the algorithm has some practical limitations (only applicable to 2 agents, requires access to environment model or transition dynamics), but still the contribution is significant. I think LOLA-based approaches are currently quite unique in trying to converge to tit-for-tat rather than unconditional cooperation that most existing methods aim for. So improvements of this approach are welcome.

The presentation has no major downsides. The appropriate literature is cited. The issue with LOLA solved by POLA is clearly formulated and validated. The algorithm is comptetently formalised and practical implementations are proposed.

The experiments are adequate, but I have a couple of concerns and propositions that I describe below.

**Minor concerns**

- In sec 2.2 it would also make sense to cite a multitude of approaches to unconditional cooperation that are not robust against defection (without details on how they work), e.g. inequity aversion, social influence, etc. This would emphasize uniqueness of LOLA.

- Lines 160-164: makes sense that POLA does not require assuming opponent’s architecture. An ablation experiment could be added to test it where the opponent’s architecture is varied (like width or depth of the opponent's network).

- Lines 232-234: why are these probabilities used to define TFT? It seems to me that having 0.64 probability to cooperate if the opponent last defected is still far from TFT. May be provide results from a different perspective by reporting in the Appendix average probabilities to cooperate in the five states for each algorithm. This way, the algorithms could be compared numerically without arbitrary thresholds.

- The coin game, while harder than IPD, is still only 3x3. Ideally, the algorithms would also be compared in environments with bigger maps (may be downscaled CleanUp or Harvest).

- In figures, linestyles and markers should be used so that all curves could be discerned without relying on colors.

---

> ### Author Response · Authors · 2022-08-01
> **Response to Reviewer kxwY**
>
> Thank you very much for the review, comments, and questions!
>
> > the algorithm has some practical limitations (only applicable to 2 agents, requires access to environment model or transition dynamics)
>
> POLA can be applied to N>2 agents. We focused on 2 agents in the paper for ease of explanation. We have added Appendix A.9 which formalizes ideal POLA for N agents and Appendix A.10 which presents N-agent POLA-DiCE. In short, agent 1 assumes all other agents $i$ find the argmin of their loss, keeping other agents’ policies fixed (using agent 1’s updated policy but the original policies of agents $j \neq 1$). We leave evaluation and further exploration to future work.
>
> Requiring access to the environment model or transition dynamics is a limitation inherited from LOLA. In principle, LOLA-based methods could learn a model of the environment first, as in model-based RL; we consider this an orthogonal improvement and leave it for future work.
>
> > but still the contribution is significant. I think LOLA-based approaches are currently quite unique in trying to converge to tit-for-tat rather than unconditional cooperation that most existing methods aim for. So improvements of this approach are welcome.
>
> Thanks for this comment - we completely agree that reciprocity-based cooperation is preferable as unconditional cooperation is easily exploited, and highlight this in our graphs where we show not just the return in self-play but also against an agent that always defects.
>
> >In sec 2.2 it would also make sense to cite a multitude of approaches to unconditional cooperation that are not robust against defection (without details on how they work), e.g. inequity aversion, social influence, etc. This would emphasize uniqueness of LOLA.
>
> Good point. Updated Section 2.2.
>
> > Lines 160-164: makes sense that POLA does not require assuming opponent’s architecture. An ablation experiment could be added to test it where the opponent’s architecture is varied (like width or depth of the opponent's network).
>
> We agree that additional experiments to support this point would be helpful, so have added Appendix B.1.6. We ran experiments in the IPD with tabular policies but varying policy parameterizations for the opponent model (tabular, neural net, pre-conditioned tabular). Even LOLA consistently finds TFT even with neural net opponent models and our original Q^i; we believe this is because the inner player policy update (usually towards defecting in each state) is less sensitive to policy parameterization than the outer player policy update which requires second order gradients and learning reciprocity. To showcase the difference between LOLA and POLA, we use new Q^i, demonstrating that POLA learns reciprocity-based cooperation whereas LOLA does not.
>
> > Lines 232-234: ... May be provide results from a different perspective by reporting in the Appendix average probabilities to cooperate in the five states for each algorithm. This way, the algorithms could be compared numerically without arbitrary thresholds.
>
> Great point. Added a table in Appendix B.1.5 reporting average probabilities of cooperation in the 5 states for all the algorithms in Figure 3. We believe the more detailed presentation strengthens our claims regarding the reliability of reciprocity-based cooperation learned by POLA.
>
> > The coin game, while harder than IPD, is still only 3x3. Ideally, the algorithms would also be compared in environments with bigger maps (may be downscaled CleanUp or Harvest).
>
> We agree that CleanUp and Harvest would be good future experiments, especially in combination with the N-agent version of POLA. To our knowledge, that would be the first time for a LOLA-related algorithm in those settings. POLA, like LOLA, requires higher-order derivatives and nobody has managed to scale algorithms from this class to larger games yet.
>
> > In figures, linestyles and markers should be used so that all curves could be discerned without relying on colors.
>
> Great suggestion - we updated figures with linestyles and markers to make curves discernible without color.
>
> > Sec 3.1: is there an explanation of why these Q^i are used? Why does LOLA fail (or why does sensitivity to parameterezation results in different equilibrium)? Does LOLA fail on other Q^i that only change basis?
>
> Section 3.1 illustrates that sensitivity to policy parameterization is the issue, and not something specific only to neural networks. The specific Q^i is just an example, which makes state CD (opponent last defected, agent cooperated) appear dissimilar from all other states. This encourages cooperating in state CD while defecting in all other states, which eventually leads to an equilibrium of always defecting. Some other Q^i that only change basis also make LOLA fail (such as the new one in Appendix B.1.6, which makes state DD dissimilar from all other states).
>
> Thank you for the detailed and helpful suggestions, comments, and questions!

---

> > ### Comment · Reviewer_kxwY · 2022-08-03
> > **Response to authors**
> >
> > Thank you for the response!
> >
> > Some of my concerns are partially alleviated, specifically regarding practical applicability. Others I now better understand as features of LOLA-based approaches rather than POLA specifically (evaluation in small environments, requiring access to transition dynamics). I also appreciate my suggestions being implemented. For these reasons, I have decided to increase my score.
> >
> > Also, I am curious. Have any prior LOLA-based approaches been extended to environments with more than two agents?

---

> > > ### Author Response · Authors · 2022-08-03
> > > **Thanks for the swift response!**
> > >
> > > Many thanks for the fast response and for increasing the score accordingly. Please feel free to let us know any follow-up questions.
> > > I am not aware of extensions of LOLA-like algorithms to more than 2 players.

---

### Official Review · Reviewer_cSUk · 2022-07-10

**Rating:** 5
**Confidence:** 3
**Soundness:** 3 good
**Presentation:** 3 good
**Contribution:** 2 fair

**Summary:**

The paper proposes POLA, a MARL agent that can learn reciprocal policies in general sum games such as iterative prisoner's dilemma (IPD). The proposed algorithm is largely based on LOLA, a MARL agent that can learn reciprocity-based cooperation in the IPD game. The main contribution of this paper is that under certain parametrization of the policy space, LOLA may fail to learn good policies. The paper highlights that a more ideal learning algorithm needs to be invariant with the policy parameterization. Although it is impractical to implement such update rules, the paper borrows ideas from PPO and proposes a proximal update version of LOLA. The proposed algorithm demonstrates better performance on IPD and the coin game.

**Questions:**

See questions in the strengths and weaknesses section.

**Limitations:**

The authors addressed the limitations and potential negative societal impact of their work.

**Strengths And Weaknesses:**

The main merit of this paper is to use a proximal update objective to improve LOLA. Although proximal policy update has been largely explored in the RL community (e.g., TRPO, PPO), it is not previously applied to LOLA. While the combination of LOLA and proximal method lead to better performance, my main concern is that the sensitivity to policy parameterization itself is not a unique problem of the social dilemma games. It is simply caused by the fact that with certain policy parameterization, even when all opponents' policy are fixed, vanilla policy gradient could fail to improve the agent's policy. If this is the case, then it should not be considered a failure of the LOLA algorithm. And this will likely limit the significance of this paper's contribution.

Additionally, while POLA demonstrates better empirical performance than LOLA in IPD and the Coin game, there are some points that I hope can be explained better either theoretically or empirically:

- The differences and connections between POLA and ideal LOLA. POLA can be viewed as the LOLA algorithm that assumes all opponents update their policies by proximal gradients, not by vanilla gradients. Will this affects POLA's ability to find reciprocity-based cooperation policy?

- In addition to the hyperparameters used by LOLA (the opponent learning rate and the self learning rate), POLA additionally includes the penalty rate \beta_in and \beta_out. Are these additional hyperparameters both crucial to the performance of POLA?

---

> ### Author Response · Authors · 2022-08-01
> **Response to Reviewer cSUk**
>
> Thank you very much for the review, comments, and questions!
>
> > my main concern is that the sensitivity to policy parameterization itself is not a unique problem of the social dilemma games. It is simply caused by the fact that with certain policy parameterization, even when all opponents' policy are fixed, vanilla policy gradient could fail to improve the agent's policy. If this is the case, then it should not be considered a failure of the LOLA algorithm. And this will likely limit the significance of this paper's contribution.
>
> With a fixed opponent policy, individual policy gradient updates may vary depending on policy parameterization, but with a sufficiently small learning rate, updates will converge to the same point in policy space. In the IPD, the optimal response to policies commonly is to always defect* and empirically we find vanilla policy gradient converges to that regardless of policy parameterization.
>
> *There are some exceptions; e.g. if the opponent policy is sufficiently close to TFT, always cooperate is optimal, but for the majority of randomly generated policies, always defect is optimal and is learned.
>
> To help understand why the issue arises with a learning opponent:
>
> 1) With a fixed opponent policy, a gradient step in the wrong direction can be corrected in subsequent updates. With a learning opponent, updates can compound as each agent’s policy updates based on the other agent’s policy. For example, suppose given the opponent’s current policy, cooperating is optimal in state CC, but the gradient update incorrectly decreases cooperation in state CC. With a fixed opponent policy, eventually future gradient updates will increase cooperation in state CC. Conversely, a learning opponent may update its policy to defect more, making it no longer optimal to cooperate in state CC, removing the chance for correction, leading to an equilibrium of always defecting.
>
> 2) Against a fixed opponent policy, finding the best response achieves the highest return. Against a learning opponent, this is no longer true; policy updates must take into account the other agent’s learning process. The opponent’s learning is also gradient-based, which introduces additional sensitivity to policy parameterization.
>
> This point is non-trivial and seems to be a key concern you have with the paper, so we want to make sure our explanation makes sense for why the issue is unique (or at least significantly more important) to the social dilemma setting. We would be happy to further clarify and discuss.
>
> Regarding the points you are looking for more explanation on:
>
> > POLA can be viewed as the LOLA algorithm that assumes all opponents update their policies by proximal gradients, not by vanilla gradients.
>
> Point of clarification: also assuming the agent itself updates by proximal optimization
>
> > Will this affects POLA's ability to find reciprocity-based cooperation policy?
>
> Yes; in a positive way, based on our empirical assessments. You correctly observe that using proximal steps is qualitatively different from gradient steps, so the policies learned by POLA may differ from LOLA even in the tabular setting. Note there is no theoretical guarantee that LOLA always finds reciprocity-based cooperation even in the tabular IPD; the demonstration is empirical, and so is ours. Empirically, we found that POLA finds reciprocity-based cooperation just as LOLA does in the tabular IPD, but much more consistently in other settings (Section 4).
>
> > In addition to the hyperparameters used by LOLA (the opponent learning rate and the self learning rate), POLA additionally includes the penalty rate \beta_in and \beta_out. Are these additional hyperparameters both crucial to the performance of POLA?
>
> Yes, these additional hyperparameters are important, particularly \beta_out (note that outer POLA doesn’t use \beta_in). We agree that introducing additional hyperparameters is a limitation of the algorithm, and state so in our limitations section. However, POLA benefits from reduced sensitivity to the learning rate (when the penalty parameters are used with multiple steps), as the magnitude of the update is controlled by the penalty, and the learning rate must be tuned only for stability and precision. For example, we note in Appendix B.1.4 that we barely tuned the outer learning rate. Overall, we believe the significant performance improvements justify the additional hyperparameters. Furthermore, in future work we can investigate adaptive penalty parameters to reduce the need to tune the penalty rates.
>
> Thank you for the great questions, suggestions and observations!

---

### Official Review · Reviewer_ghku · 2022-07-11

**Rating:** 6
**Confidence:** 3
**Soundness:** 2 fair
**Presentation:** 2 fair
**Contribution:** 2 fair

**Summary:**

This work reinterpret LOLA as approximating a proximal operator, and then derive a new algorithm called Proximal LOLA (POLA) which uses the proximal formulation directly. The updates of the POLA method are parameterization invariant, which means that behaviourally equivalent policies can result in behaviourally equivalent updates.

**Questions:**

- In the results of Full History IPD experiment (Figure 4), there is a significant degradation in the performance curve of the POLA-DiCE method. Is there any specific analysis and explanation about this phenomenon?

**Limitations:**

This work is an incremental work based on the LOLA method, simply combined with the Proximal Point Method for improvement, and does not have enough novelty.

**Strengths And Weaknesses:**

Strengths:

- This work provides good insights for improving the LOLA method in partially competitive environments.

Weakness:

- This paper is poorly written and hard to follow.
- The experiments in this paper are not solid enough. And in Section 4, the author only states the experement settings and the results, without analyzing and explaining the results of the experiment.

---

> ### Author Response · Authors · 2022-08-01
> **Response to Reviewer ghku**
>
> Thank you for the review.
>
> > This paper is poorly written and hard to follow.
>
> Could you please provide more detail on where you think the paper is poorly written and hard to follow? We put a lot of work into making the paper clear and easy to understand. For example, we have a graphic (Figure 2) explaining opponent modeling and the overall training loop for both LOLA and POLA, which goes above and beyond explanations in previous work. We would appreciate more constructive feedback so that we can improve by making changes appropriately.
>
> > The experiments in this paper are not solid enough.
>
> Could you please elaborate why you think the experiments are not solid enough, and what would make them solid? We believe our experiments convincingly support the claims set out in the paper, demonstrating the issues that arise with LOLA as a result of sensitivity to policy parameterization, and how approximations to POLA solve the issues. Our experiments cover the most common experiments in related literature (IPD, coin game), demonstrate our results with a variety of policy parameterizations including neural network function approximation, and we even include a new setting (the full history IPD) for further demonstration. We have also added further experiments in response to Reviewer kxwY’s comments in Appendix B.1.6 demonstrating that POLA allows for any choice of opponent model.
>
> > And in Section 4, the author only states the experement settings and the results, without analyzing and explaining the results of the experiment.
>
> Our intent with the experiments was to support as many claims as possible with evidence. We believe quantitative explanation through empirical results is far superior to qualitative conjecture or speculation. This may have given you the impression that we didn’t do analysis, when really, the results are directly for the purpose of analyzing, explaining, and verifying claims from previous sections.
>
> The purpose of Section 4.1 is to show that POLA learns reciprocity-based cooperation consistently across a range of policy parameterizations, whereas LOLA does so only for specific parameterizations. We already explained the effect of policy parameterization on LOLA and POLA (Figure 1, Section 3.1, Section 3.2), so did not see further need to explain again in Section 4.1.
>
> Subsequent experiments (Sections 4.2, 4.3) further verify our previous claims. We go beyond showing just the return in self-play (as is common in related work), explicitly providing extra plots versus an opponent who always defects, as analysis showing that the policies learned are truly reciprocity-based, not unconditional cooperation.
>
> Of course, there could be places where we could have better explained certain concepts or specific surprising results, so we would appreciate more specific feedback (for example, like your question below, as well as specific comments and questions made by other reviewers).
>
> > In the results of Full History IPD experiment (Figure 4), there is a significant degradation in the performance curve of the POLA-DiCE method. Is there any specific analysis and explanation about this phenomenon?
>
> Respectfully, we disagree that there is a significant degradation in the performance curve of POLA-DiCE in Figure 4. The error bars show significant overlap with the POLA-OM results and remain very close to the maximum (always cooperate) reward.
>
> The need to use samples and rollouts in the Full History IPD and coin game settings inherently results in noise, which can be amplified by the approximation to ideal POLA which POLA-DiCE provides. Without the exact value function, and with limited batch size, we expect noise to degrade performance in some experiment runs; this is why the error bars are fairly large.
>
> > This work is an incremental work based on the LOLA method, simply combined with the Proximal Point Method for improvement, and does not have enough novelty.
>
> “Simply combined with the Proximal Point Method” greatly undersells our contribution. We identify and demonstrate a problem with LOLA that wasn’t previously known, conceptually reinterpret LOLA as approximating a proximal operator, show that our proximal formulation is well-suited to address the issue, and develop and evaluate practical approximations.
>
> Overall, we would appreciate more detailed and specific comments, which would let us better discuss and address your concerns, as well as help us improve our work.
>
> We thank you for taking the task of reviewing seriously and look forward to a more constructive response at your earliest convenience.

---

> > ### Comment · Reviewer_ghku · 2022-08-07
> > **Thank you for your reply.**
> >
> > The concerns have been partially addressed, so I will consider raising my score. However, I insist that the writing and presentation of this paper is a little obscure and some readers may not be able to grasp its ideas and logic quickly. This is an important criterion for papers to be published in a top conference like NeurIPS. I suggest that the authors could organize the connections between the sections more tightly and highlight the motivation and contributions.

---

> > > ### Author Response · Authors · 2022-08-08
> > > **Added Section to Highlight Motivations and Contributions and Clarify Connections between Sections**
> > >
> > > Thank you for your response and suggestions! We have added Section 1.1 to highlight our motivation and contributions. We also hope this makes the connections between subsequent sections more clear.
> > >
> > > We understand that there is a lot of information in the paper, and despite our best efforts to be clear and understandable, due to limited space, our presentation may be dense or terse in certain areas. If there are specific parts of the paper you found obscure or difficult to understand, we would appreciate further detailed feedback so we could better explain. Thank you for helping us improve the paper.

---

### Meta-Review · Area_Chair_ygRw · 2022-08-26

**Recommendation:** Accept
**Confidence:** Less certain

**Metareview:**

I read the paper, review, and responses. I'm not an expert in this sub-area. I'm consider it an incremental work with borderline accept/reject evaluation based on the paper and reviews. OK with an accept.

**Award:**

No

---

### Decision · Program_Chairs · 2022-09-14

Accept